# Factorized-FL: Personalized Federated Learning with Parameter Factorization & Similarity Matching

**Wonyong Jeong**
Graduate School of AI
KAIST, Seoul, South Korea
wyjeong@kaist.ac.kr

**Sung Ju Hwang**
Graduate School of AI
KAIST, Seoul, South Korea
sjhwang82@kaist.ac.kr

## Abstract

In real-world federated learning scenarios, participants could have their own personalized labels incompatible with those from other clients, due to using different label permutations or tackling completely different tasks or domains. However, most existing FL approaches cannot effectively tackle such extremely heterogeneous scenarios since they often assume that (1) all participants use a synchronized set of labels, and (2) they train on the same tasks from the same domain. In this work, to tackle these challenges, we introduce `Factorized-FL`, which allows to effectively tackle label- and task-heterogeneous federated learning settings by factorizing the model parameters into a pair of rank-1 vectors, where one captures the common knowledge across different labels and tasks and the other captures knowledge specific to the task for each local model. Moreover, based on the distance in the client-specific vector space, `Factorized-FL` performs a selective aggregation scheme to utilize only the knowledge from the relevant participants for each client. We extensively validate our method on both label- and domain-heterogeneous settings, on which it outperforms the state-of-the-art personalized federated learning methods. The code is available at https://github.com/wyjeong/Factorized-FL.

## 1  Introduction

Personalized Federated Learning (PFL) aims to utilize the aggregated knowledge from other clients while learning a client-specific model that is specialized for its own task and data distribution, rather than learning a universal, global model for tackling all local tasks [1, 16, 5, 28]. While various personalized federated learning approaches have shown success in alleviating the data heterogeneity problem, yet, they are also limited as they follow the common assumptions of the standard federated learning setting, that (1) all participants use a synchronized set of labels, and (2) all clients tackle the same task from the same domain.

In many real-world scenarios, the first assumption may not hold since the labels for the same task could be differently annotated depending on the user environment, i.e. language or nationality (Figure 2 Left). In other words, when working with the same set of semantic classes, the labels across multiple clients could be completely different. For example, the same "Car" class may be given the label "Vehicle" or "Auto".

The second assumption severely limits the pool of devices or institutions that can participate in the collaborative learning process. However, clients working on different tasks and domains may have similar classes, or a common underlying knowledge, that may be helpful for the local models being trained at other clients (Figure 2 Right). Thus, it would be helpful if we can allow the local models with heterogeneous tasks and domains to communicate the common knowledge across them.

36th Conference on Neural Information Processing Systems (NeurIPS 2022).

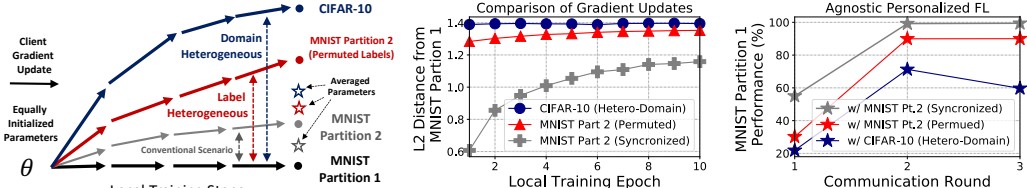

| (a) Illustration of the Parameter Space | (b) Distance of Gradient Updates | (c) Performance Degeneration |

Figure 1: **Challenges of Label & Domain Heterogeneous Scenarios** (a) illustrates label and domain heterogeneity in parameter space. (b) shows normalized $L_2$ distance of the gradient updates from that of model trained on MNIST Partition 1. (c) shows performance degradation on MNIST partition 1 caused by the label and domain heterogeneity while performing federated learning.

However, federated learning under label and domain heterogeneity is a nontrivial problem, as most methods suffer from severe performance degeneration in such settings (Table 1). We analyze this phenomenon in Figure 1. Specifically, we train equally-initialized models on four different datasets and observe how different the gradient updates become as training goes on (we measure normalized $L_2$ distance between them). Learning on two MNIST partitions will yield the smallest difference between gradients (Figure 1 (a) and (b) Gray). Interestingly, simply permuting the labels of the partition 2 makes the

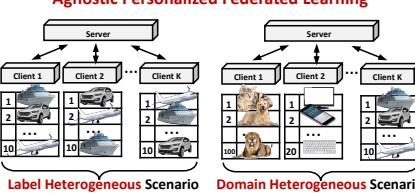

Figure 2: **Label and Domain Heterogeneous FL Scenarios. Left** labels are not synchronized across all clients. **Right** the local clients learn on different tasks and/or domains.

gradient updates to largely diverge from the original gradients, compared to those of the model trained with synchronized labels (Figure 1 (a) and (b) Red). Moreover, learning on the completely different dataset (CIFAR-10) makes model gradients diverge more severely compared to learning on the dataset with permuted labels (Figure 1 (a) and (b) Blue).

The heterogeneity in the labels and domains tackled by each participant leads to severe performance degeneration when performing federated learning (Figure 1 (c)). We measure performance on MNIST partition 1 while aggregating a model trained on the different dataset for every 3 epochs. We observe averaged models suffer from crucial performance degeneration in both FL scenarios. Note that the classifier layers are not shared when performing federated learning. However, such label- and domain-heterogeneous FL settings have been relatively overlooked. Although few previous works [1, 16, 25] briefly introduced such extreme settings as motivations, they did not formally define the problems nor tackle such scenarios, and rather targeted standard non-IID scenarios.

We name this challenging problem as the Agnostic Personalized Federated Learning (APFL) problem, where participants with personalized labels or from multiple domains can collaboratively learn while benefiting each other. An APFL problem has two critical challenges: (1) Label Heterogeneity for the discrepancy of the labels, due to the lack of a synchronized labeling scheme across the clients; and (2) Domain Heterogeneity for the discrepancy in the task and domains tackled by each participant. To tackle these challenges, we propose a novel FL framework, dubbed `Factorized-FL`, which factorizes model parameters into basis vectors and aggregate them in the factorized parameter space. This allows to factorize the model aggregation to take place in a semantic parameter basis space which is more robust to the use of different labels and data. Also, the factorization results in the separation of the client-general and client-specific knowledge, and thus prevents the aggregation of incompatible knowledge across clients. Moreover, to further alleviate the model from collapsing into a degenerate solution, we measure the task similarity across the clients using the factorized parameters, to allow selective aggregation of the knowledge among the relevant models that work on similar tasks or domains. We extensively validate our method on both label- and domain-heterogeneous settings, and show that it significantly outperforms the current state-of-the-art methods with lower communication cost due to using factorized parameters. Our contributions can be summarized as follows:

- We introduce the Agnostic Personalized Federated Learning (APFL) problem and study its two critical challenges, Label and Domain Heterogeneity.
- We propose Factorized-FL for tackling the APFL problem, which factorizes model parameters to reduce parameter dimensionality for alleviating knowledge collapse, and utilize them to measure task-level similarity for matching relevant clients.
- We validate our method in both label- and domain-heterogeneous scenarios and show our method outperforms the current state-of-the-art personalized federated learning methods.

## 2    Related Work

**Federated learning**    Many federated learning algorithms have been proposed since the introduction of `FedAvg` [18], but, we specifically focus on works that aim to tackle the heterogeneity problems, e.g. Non-IIDness. Some studies focus on regularization methods [19, 15], correcting disparity between server and clients [26, 9], or contrastive learning [13]. Recently, many existing works also tackle architecture-level heterogeneity [24, 29, 3].

**Personalized federated learning** aims to improve the individual local models instead of the universal global model, via the mixture methods [17, 2, 7], meta-learning [5], partial network aggregation [1, 16], or hyper-networks [25]. Recent approaches aim to enhance personalization performance by avoiding to aggregate irrelevant knowledge that are not helpful. Zhang et al. [28] download the models from other clients to aggregate only the beneficial ones for the local task at each client. Sattler et al. [23], Duan et al. [4] measure client-wise similarity by using the gradient updates. Our method also measures client similarity but in a more efficient way, by utilizing 1D vectors from factorization.

**Re-parameterization for federated learning** Jeong et al. [8], Yoon et al. [27] decompose model parameters (using an additional set of parameters) to train them with different objectives, which do not reduce the dimensionality of model parameters. Some approaches factorize high dimensional model parameters into low rank matrices.  Konečnỳ et al. [10] introduce structured update which model directly learns factorized parameter space, and  Nam et al. [20] propose to use the Hadamard product of low rank matrices to enhance communication efficiency. Unlike prior works, we utilize rank-1 vectors factorization to tackle heterogeneity in domains and tasks, by allowing the 1D vectors to capture task-general and the client-specific knowledge. Also, we minimize the information loss from factorization by utilizing sparse bias matrices.

## 3    Problem Definition

We begin with the formal definition of the conventional federated learning scenario, and then introduce our novel Agnostic Personalized Federated Learning (APFL) problem.

### 3.1    Preliminaries

Our goal is to solve a multi-class classification task under an FL setting. Let $f_g$ be a global model (neural network) at the global server and $\mathcal{F} = \{f_k\}_{k=1}^K$ be a set of $K$ local neural networks, where $K$ is the number of local clients. $\mathcal{D} = \{\mathbf{x}_i, y_i\}_{i=1}^N$ be a given dataset, where $N$ is the number of instances, $\mathbf{x}_i \in \mathbb{R}^{W \times H \times D}$ is the $i_{th}$ examples in a size of width $W$, height $H$, and depth $D$, with a corresponding target label $y_i \in \{1, \ldots, C\}$ for the $C$-way multi-class classification problem. The given dataset $\mathcal{D}$ is then disjointly split into $K$ sub-partitions $\mathcal{D}_k = \{\mathbf{x}_{k,i}, y_{k,i}\}_{i=1}^{N_k}$ s.t. $\mathcal{D} = \bigcup_{k=1}^K \mathcal{D}_k$, which are distributed to the corresponding local model $f_k$. Let $R$ be the total number of the communication rounds and $r$ denote the index of the $r_{th}$ communication round. At the initial round $r{=}1$, the global model $f_g$ initializes the global weights $\theta_{f_g}^{(1)}$ and broadcasts $\theta_{f_g}^{(1)}$ to an arbitrary subset of local models that are available for training at round $r$, such that $\mathcal{F}^{(r)} \subset \mathcal{F}$, $|\mathcal{F}^{(r)}| = K^{(r)}$, and $K^{(r)} \leq K$, where $K^{(r)}$ is the number of available local models at round $r$. Then the active local models $f_k \in \mathcal{F}^{(r)}$ perform local training to minimize loss $\mathcal{L}(\theta_k^{(r)})$ on the corresponding sub-partition $\mathcal{P}_k$ and update their local weights $\theta_k^{(r+1)} \leftarrow \theta_k^{(r)} - \eta \nabla \mathcal{L}(\theta_k^{(r)})$, where $\theta_k^{(r)}$ is the set of weights for the local model $f_k$ at round $r$ and $\mathcal{L}(\cdot)$ is the loss function. When the local training is done, the global model $F$ collects and aggregates the learned weights $\theta_{f_g}^{(r+1)} \leftarrow \frac{N_k}{N} \sum_{i=1}^{K^{(r)}} \theta_k^{(r)}$ and then broadcasts newly updated weights to the local models available at the next round $r + 1$. These learning procedures are repeated until the final round $R$.

On the other hand, Personalized Federated Learning aims to adapt the individual local models $f_{1:K}$ to their local data distribution $\mathcal{D}_{1:K}$, to obtain specialized solution for each task at the local client, while utilizing the knowledge from other clients. Thus merging the local knowledge for personalized FL is not necessarily done in the form of $\theta_{f_g}^{(r+1)} \leftarrow \frac{N_k}{N} \sum_{i=1}^{K^{(r)}} \theta_k^{(r)}$, and the specific ways to utilize the knowledge from others depends on the specific algorithm, i.e. $\theta_k^{(r+1)} \leftarrow \theta_k^{(r)} + \sum_{i \neq k}^{K^{(r)}} \omega_i(\theta_k^{(r)} - \theta_i^{(r)})$, where $\omega(\cdot)$ is weighing function  [28].

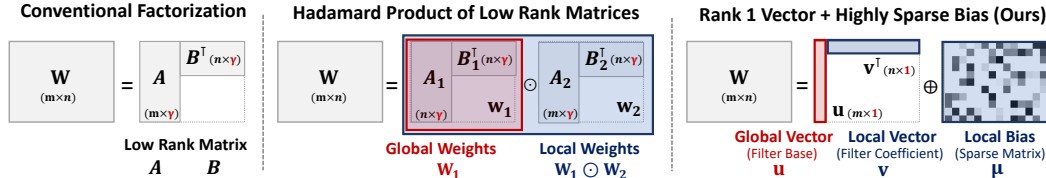

Figure 3: **Illustration of Parameter Factorization Methods:** Left shows conventional matrix factorization with two low rank matrices with rank $\gamma$. Middle represents the method utilizing Hadamard product of low rank matrix for federated learning [20]. Right illustrates our factorization method for agnostic personalized federated learning, which utilizes rank 1 vectors and highly sparse bias.

## 3.2 Agnostic Personalized Federated Learning

Agnostic Personalized Federated Learning (APFL) is a scenario where any local participants from diverse domains with their own personalized labeling schemes can collaboratively learn, benefiting each other. There exist two critical challenges that need to be tackled to achieve this objective: (1) Label Heterogeneity and (2) Domain Heterogeneity.

**Label Heterogeneity**     This scenario assumes that the labeling schemes are not perfectly synchronized across all clients, as described in Section 1 and Figure 2 Left. Most underlying setting for this scenario is the same as the conventional single-domain setting with synchronized labels that is described in Section 3.1, except that labels are arbitrarily permuted amongst clients. The local data $\mathcal{D}_k$ for the local model $f_k$ is now defined as $\mathcal{D}_k = \{\mathbf{x}_{k,i}, \varphi_k(y_{k,i})\}_{i=1}^{N_k}$, where $\varphi_k(\cdot)$ is a mapping function for the local model $f_k$ which maps a given class $y_{k,i}$ with a randomly permuted label $p_{k,i} = \varphi_k(y_{k,i})$. Let the $j$th layer out of $L$ layers in the neural networks of local model $f_k$ be $\ell_k^j$ and the last layer $\ell_k^L$ be the classifier layer. Since each client has differently permuted labels, the personalized classifiers $\ell_{1:K}^L$ are no longer compatible to each other. While we can merge the layers below the classifier in this setting, training with heterogeneous labels could still lead to large disparity in the local gradients even in the initial communication round, as described in Figure 1.

**Domain Heterogeneity**     This scenario presumes that local clients learn on their own dataset $\mathcal{D}$, that are completely different from the datasets that are used at other clients, as described in Section 1 and Figure 2 Right. In this setting, $K$ disjoint datasets $\mathcal{D}_{1:K}$ are assigned to the $K$ local clients $f_{1:K}$, where $\mathcal{D}_k = \{\mathbf{x}_{k,i}, y_{k,i}\}_{i=1}^{N_k}$ is the dataset assigned to the local model $f_k$. The number of target classes may differ across clients, such that $y_{k,i} \in \{1, \ldots, C_k\}$. We assume complete disjointness across clients, such that there is no instance- and class-wise overlap across the datasets: $\varnothing = \bigcap_{k=1}^{K} \mathcal{D}_k$ and $\varnothing = \bigcap_{k=1}^{K} \mathcal{C}_k$, where $\mathcal{C}_k$ is a set of classes for client $k$. Similarly to the label-heterogeneous scenario, the personalized classifiers $\ell_{1:K}^L$ are no longer compatible to each other due to the heterogeneity in the data and the labels. Hence, the aggregation is done for the layers before the classifier, but they will be also incompatible as the learned model weights will be largely different across domains.

## 4 Factorized Federated Learning

### 4.1 Factorization of Model Parameters

Wang et al. [26] discussed that the conventional knowledge aggregation, that is often performed in a coordinate-wise manner, may have severe detrimental effects on the averaged model. This is because the deep neural networks have extremely high-dimensional parameters and thus meaningful element-wise neural matching is not guaranteed when aggregating the weights across different models trained under diverse settings.

One naive solution to this problem is to factorize model parameters into lower dimensional space, i.e. low rank matrices, as shown in Figure 3 (Left). Conventional approaches, such as SVD, Tucker, or Canonical Polyadic decomposition, however, factorize model parameters after training [12, 22] is done. Thus, the dimensionality at the time of knowledge aggregation will remain the same as the unfactorized model. Konečnỳ et al. [10], Nam et al. [20] pre-decompose model parameters to low rank matrices for FL scenarios. While Konečnỳ et al. [10] use naive low rank matrices, Nam et al. [20] use two sets of low rank matrices to improve expressiveness and utilize them as global and local weights (Figure 3 (Middle)). Unlike prior works, our approach utilizes rank-1 vectors to

perform aggregation in the lowest subspace possible for compatibility, while effectively yet efficiently enhancing expressiveness with sparse bias matrices, as shown in Figure 3 (Right) and Figure 5 (Left). Another crucial difference of our method from the previous factorization methods is that our rank-1 vectors have distinct roles. Our factorization will separate the common knowledge from the task- or domain-specific knowledge, since $\mathbf{u}$ could be thought as the bases (the common knowledge across clients) and $\mathbf{v}$ could be thought as the coefficients (client-specific information).

In Figure 4 (a) and (b), which show the experimental results with the factorized model, we observe that $\mathbf{u}$ trained on two datasets becomes closer to each other (MNIST Partition 1) while $\mathbf{v}$ (personalized filter coefficient) remain largely different as FL goes on. With this observation, we further aggregate $\mathbf{u}$ while allowing $\mathbf{v}$ to be different across clients, to allow personalized FL. Further, we use the client specific $\mathbf{v}$ for similarity matching, to identify relevant local models from other clients. In the following paragraphs, we describe our factorization method in detail, for both fully-connected and convolutional layers.

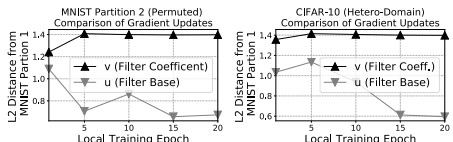

(a) Hetero. Labels    (b) Hetero. Domains
Figure 4: **Analysis of u and v:** The normalized $L_2$ distance of the gradient updates of $\mathbf{u}$ and $\mathbf{v}$ while learning on (a) MNIST Partition 2 and (b) CIFAR-10 compared to learning on MNIST Partition 1.

**Factorization of Fully-Connected Layers**    We assume that each local model $f_k$ has a set of local weights $\theta_k$ across all layers; that is, $\theta_k = \{\mathbf{W}_k^i\}_{i=1}^L$. The dimensionality of the dense weight $\mathbf{W}_k^i$ for each fully connected layer is $\mathbf{W}_k^i \in \mathbb{R}^{I \times O}$, where $I$ and $O$ indicate respective input and output dimensions. We can reduce the $I \times O$ complexity by factorizing the high order matrix into the outer product of two vectors as follows:

$$\mathbf{W}_k^i = \mathbf{u}_k^i \times \mathbf{v}_k^{i\mathsf{T}}, \text{where } \mathbf{u}_k^i \in \mathbb{R}^I, \mathbf{v}_k^i \in \mathbb{R}^O. \tag{1}$$

However, such extreme factorization of the weight matrices may result in the loss of expressiveness in the parameter space. Thus, we additionally introduce a highly sparse bias matrix $\mu$ to further capture the information not captured by the outer product of the two vectors as follows:

$$\mathbf{W}_k^i = \mathbf{u}_k^i \times \mathbf{v}_k^{i\mathsf{T}} \oplus \mu_k^i, \text{where } \mathbf{u}_k^i \in \mathbb{R}^I, \mathbf{v}_k^i \in \mathbb{R}^O, \mu_k^i \in \mathbb{R}^{I \times O}. \tag{2}$$

We initialize $\mu$ with zeros so that it can gradually capture the additional expressiveness that are not captured by $\mathbf{u}$ and $\mathbf{v}$ during training. We can control its sparsity by the hyper-parameter for the sparsity regularizer described in 4.3.

**Factorization of Convolutional Layers**    The difference between the fully-connected and convolutional layers is that the convolutional layers have multiple kernels (or filters) such that $\mathbf{W}_k^i \in \mathbb{R}^{F \times F \times I \times O}$, where $F$ is a size of filters (we assume the filter size is equally paired for the simplicity). To induce $\mathbf{u}$ to capture base filter knowledge and $\mathbf{v}$ to learn filter coefficient, it is essential to design $\mathbf{u} \in \mathbb{R}^{F \cdot F}$ and $\mathbf{v} \in \mathbb{R}^{I \cdot O}$, but not in arbitrary ways, such as $\mathbf{u} \in \mathbb{R}^{I \cdot F}$ and $\mathbf{v} \in \mathbb{R}^{O \cdot F}$ or $\mathbf{u} \in \mathbb{R}^O$ and $\mathbf{v} \in \mathbb{R}^{I \cdot F \cdot F}$. We observe that performance is degenerated when the parameters are ambiguously factorized (Figure 7 (h)). Our proposed factorization method for convolutional layers are as follows:

$$\mathbf{W}_k^i = \pi(\mathbf{u}_k^i \times \mathbf{v}_k^{i\mathsf{T}} \oplus \mu_k^i), \text{where } \mathbf{u}_k^i \in \mathbb{R}^{F \cdot F}, \mathbf{v}_k^i \in \mathbb{R}^{I \cdot O},$$
$$\mu_k^i \in \mathbb{R}^{F \cdot F \times I \cdot O}, \pi(\cdot) : \mathbb{R}^{F \cdot F \times I \cdot O} \to \mathbb{R}^{F \times F \times I \times O}, \tag{3}$$

$\pi(\cdot)$ is the weight reshaping function. Note that we reparameterize our model *at initialization time*. Then we reconstruct and train full weights of each layer $\mathbf{W}_k^{1:L}$, while optimizing $\mathbf{u}_k^{1:L}$, $\mathbf{v}_k^{1:L}$, and $\mu_k^{1:L}$, respectively, during training phase.

## 4.2 Similarity Matching

Since we assume label- and domain-heterogeneous FL scenarios, aggregating the parameter bases across all clients may not be optimal, since some of them could be highly irrelevant. Yoon et al. [27] and Zhang et al. [28] also demonstrate that avoiding aggregation of irrelevant models from other clients improves local model performance. Yoon et al. [27] achieve this goal by taking the weighted combination of task-specific weights from other clients, and Zhang et al. [28] suggest downloading the models from other clients and evaluating their performance on a local validation set, at each client. However, since they require additional communication and computing cost at the local clients, we provide a more efficient yet effective approach to match models that are beneficial to each other.

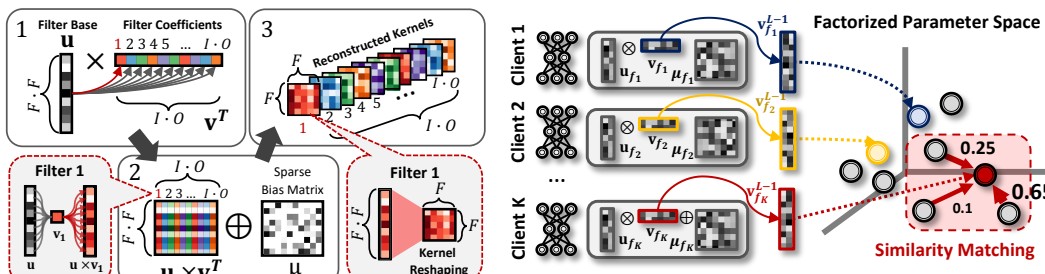

(a) Factorization for Base and Specific Knowledge      (b) Similarity Matching on Factorized Space

Figure 5: **Illustration of the parameter factorization & similarity matching algorithm (a)** We multiply two factorized vectors $\mathbf{u}$ and $\mathbf{v}$ to obtain kernel matrix. Then we add the sparse bias matrix $\mu$ to complement non-linearity and reshape the matrix into the original kernel shape. **(b)** We match relevant clients utilizing the factorized vector $\mathbf{v}$ that captures client-specific knowledge. Then we aggregate $\mathbf{u}$ based on the similarity.

**Efficient similarity matching** Our method utilizes a factorized vector $\mathbf{v}_k$ for measuring similarity across different models, at the central server. Since $\mathbf{v}$ are devised to learn personalized coefficients, we assume that clients trained on similar tasks or domains will have similar $\mathbf{v}$. Specifically, we use $\mathbf{v}_k^{L-1}$ of the layer before the classifier layer for similarity matching. The similarity matching function $\Omega(\cdot)$, is defined as the cosine similarity between target client $f_k$ and the other clients $\{f_i\}_{i \neq k}^K$, as follows:

$$\Omega(\mathbf{v}_{f_k}^{L-1}, \mathbf{v}_{f_{i \neq k:K}}^{L-1}) = \{\sigma_i | \sigma_i = \frac{\mathbf{v}_{f_k}^{L-1} \cdot \mathbf{v}_{f_i}^{L-1}}{\|\mathbf{v}_{f_k}^{L-1}\| \|\mathbf{v}_{f_i}^{L-1}\|}, \sigma_i \geq \tau\}_{i \neq k}^K. \tag{4}$$

The similarity scores for those with the cosine similarity scores lower than the given threshold $\tau$, are set to zero. Our method is significantly more efficient than the similarity matching approaches which use full gradient updates for clustering clients [23, 4], as we only use single vector $\mathbf{v}_k^{L-1}$ from each client $k$ that are obtained without additional cost.

**Personalized weighted averaging** We allow each local model to individually perform weighted aggregation of the model weights from other clients, utilizing their similarity scores:

$$\mathbf{u}_k^l \leftarrow \frac{\exp(\epsilon \cdot \sigma_i)}{\sum_{i=1}^K \exp(\epsilon \cdot \sigma_i)} \sum_{i=1}^K \mathbf{u}_i^l, s.t. \forall l \in \{1, 2, \ldots, L-1\}. \tag{5}$$

where $\epsilon$ is a hyperparameter for scaling the similarity score $\sigma_i$. We always set $\sigma_k$, the similarity score for itself, as $1.0$. We name this model that aggregates only basis vector $\mathbf{u}_k$ from each client $k$ as `Factorized-FL` $\alpha$. This will result in encoding common knowledge across local models in $\mathbf{u}_k$ which is especially suitable for domain-heterogeneous scenarios. However, for label-heterogeneous scenarios, we empirically observed that aggregating all parameters results in better performance, as the factorization helps averaging them in a more compatible space. We thus introduce an additional variant, namely `Factorized-FL` $\beta$, which aggregates not only the parameter bases across all clients, but also coefficient and bias terms as well. Yet, note that `Factorized-FL` $\beta$ require significantly larger communication overhead over `Factorized-FL` $\alpha$.

### 4.3 Learning Objective

Now we describe our final learning objective. Instead of utilizing the single term $\theta_k$ for local weights of neural network $f_k$, now let $\mathcal{U}_k$, $\mathcal{V}_k$, and $\mathcal{M}_k$ be sets of $\mathbf{u}_k$, $\mathbf{v}_k$, and $\mu_k$ of all layers in $f_k$, s.t. $\mathcal{U}_k = \{\mathbf{u}_k^i\}_{i=1}^L$, $\mathcal{V}_k = \{\mathbf{v}_k^i\}_{i=1}^L$, and $\mathcal{M}_k = \{\mu_k^i\}_{i=1}^L$, then our local objective function is,

$$\min_{\mathcal{U}_k, \mathcal{V}_k, \mathcal{M}_k} \sum_{\mathcal{B} \in \mathcal{D}_k} \mathcal{L}(\mathcal{B}; \mathcal{U}_k, \mathcal{V}_k, \mathcal{M}_k) + \lambda_{\text{sparsity}} \|\mathcal{M}_k\|_1, \tag{6}$$

where $\mathcal{L}$ is the standard cross-entropy loss performed on all minibatch $\mathcal{B} \in \mathcal{D}_k$. We add the $L_1$ sparsity inducing regularization term to make the bias parameters highly sparse, controlling its effect with a hyperparameter $\lambda_{\text{sparsity}}$. As for the full training procedure, please see the pseudo-code of the algorithm in the supplementary file (Section A).

Table 1: **Performance comparison of label and domain heterogeneous scenario. Top (label heterogeneous scenario):** we train 20 clients on each dataset (CIFAR-10 & SVHN) for 250 training iterations ($E$=5, $R$=50). **Bottom (domain & label heterogeneous scenario):** We train 20 clients for 500 training iterations ($E$=5,$R$=100) on 20 sub-datasets from 5 heterogeneous domains (4 clients per domain). Labels are also permuted for all partitions. We measure averaged performance over three trials with different seeds. (Table 1 Bottom will be moved to Appendix and a new set of experiments for the pure domain heterogeneous setting will be added.)

| Dataset | Method | Standard IID | | Permuted IID | | Standard Non-IID | | Permuted Non-IID | |
|---|---|---|---|---|---|---|---|---|---|
| | | Accuracy [%] | Cost [Gb] | Accuracy [%] | Cost [Gb] | Accuracy [%] | Cost [Gb] | Accuracy [%] | Cost [Gb] |
| CIFAR-10 | Stand-Alone | 64.31 (± 1.08) | - | 63.93 (± 0.90) | - | 47.79 (± 0.91) | - | 46.06 (± 1.03) | - |
| | FedAvg [18] | 70.28 (± 0.82) | 20.39 | 65.31 (± 1.28) | 20.39 | 53.08 (± 1.4) | 20.39 | 48.90 (± 1.25) | 20.39 |
| | FedProx [14] | 70.54 (± 0.73) | 20.39 | 66.28 (± 0.90) | 20.39 | 53.56 (± 0.55) | 20.39 | 47.86 (± 0.83) | 20.39 |
| | Clustered-FL [23] | 69.48 (± 1.02) | 20.39 | 65.77 (± 1.03) | 20.39 | 53.93 (± 1.57) | 20.39 | 49.00 (± 0.32) | 20.39 |
| | Per-FedAvg [5] | 70.84 (± 1.01) | 20.39 | 65.58 (± 0.74) | 20.39 | 53.35 (± 2.87) | 20.39 | 47.60 (± 1.01) | 20.39 |
| | FedFOMO [28] | 70.19 (± 0.79) | 122.33 | 64.26 (± 0.92) | 122.33 | 50.69 (± 1.61) | 122.33 | 46.73 (± 1.04) | 122.33 |
| | pFedPara [20] | 67.96 (± 1.25) | 7.4 | 65.12 (± 1.27) | 7.4 | 55.88 (± 1.28) | 7.4 | 50.22 (± 0.92) | 7.4 |
| | Factorized-FL $\alpha$ (Ours) | 66.97 (± 1.36) | **0.32** | 67.91 (± 1.08) | **0.32** | 50.34 (± 1.33) | **0.32** | 50.24 (± 1.03) | **0.32** |
| | Factorized-FL $\beta$ (Ours) | **76.26** (± **1.05**) | 18.25 | **70.59** (± **2.07**) | 18.25 | **65.30** (± **1.38**) | 18.25 | **56.61** (± **1.10**) | 18.25 |
| SVHN | Stand-Alone | 84.18 (± 0.37) | - | 84.32 (± 0.31) | - | 62.50 (± 0.84) | - | 62.11 (± 0.78) | - |
| | FedAvg [18] | 88.53 (± 0.32) | 20.39 | 87.83 (± 0.29) | 20.39 | 76.03 (± 0.90) | 20.39 | 69.73 (± 0.91) | 20.39 |
| | FedProx [14] | 89.04 (± 0.33) | 20.39 | 87.31 (± 0.21) | 20.39 | 76.61 (± 0.92) | 20.39 | 69.40 (± 0.73) | 20.39 |
| | Clustered-FL [23] | 88.02 (± 0.37) | 20.39 | 87.33 (± 0.29) | 20.39 | 74.27 (± 0.83) | 20.39 | 68.84 (± 0.84) | 20.39 |
| | Per-FedAvg [5] | 88.46 (± 0.53) | 20.39 | 87.29 (± 0.24) | 20.39 | 74.90 (± 0.58) | 20.39 | 68.67 (± 0.79) | 20.39 |
| | FedFOMO [28] | 88.34 (± 0.26) | 122.33 | 84.03 (± 0.34) | 122.33 | 72.12 (± 0.96) | 122.33 | 61.45 (± 0.93) | 122.33 |
| | pFedPara [20] | 88.70 (± 0.25) | 7.4 | 88.24 (± 0.22) | 7.4 | 75.36 (± 0.93) | 7.4 | 70.26 (± 0.85) | 7.4 |
| | Factorized-FL $\alpha$ (Ours) | 86.56 (± 0.39) | **0.32** | 86.31 (± 0.27) | **0.32** | 66.25 (± 0.71) | **0.32** | 66.12 (± 0.79) | **0.32** |
| | Factorized-FL $\beta$ (Ours) | **91.04** (± **0.73**) | 18.25 | **89.57** (± **0.47**) | 18.25 | **81.07** (± **0.53**) | 18.25 | **74.63** (± **0.84**) | 18.25 |

| Method | Household | Fruit&Food | Tree&Flower | Transport | Animals | AVERAGE | |
|---|---|---|---|---|---|---|---|
| | Accuracy [%] | Accuracy [%] | Accuracy [%] | Accuracy [%] | Accuracy [%] | Accuracy [%] | Cost [Gb] |
| Stand-Alone | 59.38 (± 0.70) | 63.74 (± 1.76) | 61.20 (± 0.64) | 63.22 (± 2.12) | 58.40 (± 1.20) | 61.35 (± 1.90) | - |
| FedAvg [18] | 55.08 (± 2.49) | 63.18 (± 2.45) | 57.76 (± 1.77) | 57.96 (± 2.97) | 53.61 (± 1.21) | 56.42 (± 1.65) | 40.78 |
| FedProx [14] | 56.77 (± 2.59) | 61.33 (± 1.28) | 58.14 (± 0.51) | 55.79 (± 0.82) | 51.43 (± 2.17) | 56.71 (± 1.52) | 40.78 |
| Clustered-FL [23] | 59.44 (± 2.31) | 66.93 (± 0.88) | 60.03 (± 1.13) | 62.17 (± 2.55) | 55.01 (± 2.14) | 59.20 (± 2.16) | 40.78 |
| Per-FedAvg [5] | 64.01 (± 1.56) | 67.68 (± 1.07) | 61.62 (± 1.86) | 64.36 (± 1.27) | 60.25 (± 0.88) | 62.92 (± 1.60) | 40.78 |
| FedFOMO [28] | 59.70 (± 1.78) | 64.32 (± 1.48) | 63.87 (± 2.19) | 62.57 (± 0.97) | 57.75 (± 2.28) | 62.07 (± 1.80) | 244.66 |
| pFedPara [20] | 60.35 (± 3.30) | 65.56 (± 0.60) | 61.98 (± 2.02) | 60.16 (± 6.66) | 56.12 (± 2.86) | 61.11 (± 2.61) | 15.98 |
| Factorized-FL $\alpha$ (Ours) | 64.06 (± 0.16) | **68.55** (± **0.16**) | 64.39 (± 2.23) | **66.93** (± **1.03**) | 61.33 (± 3.56) | **64.49** (± **1.57**) | **0.64** |
| Factorized-FL $\beta$ (Ours) | **65.04** (± **0.78**) | 65.53 (± 2.79) | **65.14** (± **2.64**) | 65.04 (± 1.95) | **64.45** (± **3.47**) | 63.93 (± 2.26) | 36.5 |

Figure 6: **Test Accuracy Curves & Communication Costs** We plot the averaged test accuracy curves on both CIFAR-10 and SVHN for (a) IID, (B) Permuted IID, (c) Non-IID, and (d) Permuted Non-IID partitions. For (e), we plot test accuracy curves (top) and communication costs (bottom) for domain & label heterogeneous dataset. All plots for communication costs are represented in the supplementary material.

# 5 Experiment

## 5.1 Experimental Setup

**Baselines** We first consider well-known baseline FL methods, such as (1) FedAvg [18] and (2) FedProx [14]. We evaluate our factorization technique with (3) pFedPara [20] which also uses kernel factorization technique for personalized FL scenarios. Our similarity matching approach is compared to (4) Clustered-FL [23] and (5) FedFOMO [28], which measuring client-wise similarity or helpfulness. (6) Per-FedAvg [5] is also used for evaluation as it shows great performance on heterogeneous federated learning scenarios. We also show local training model, (7) Stand-Alone, for the lower bound performance. Detailed information are described in Appendix (Section C).

**Datasets** (1) **Label Heterogeneous Scenario**: we use CIFAR-10 [11] and SVHN [21] datasets and create four different partitions for each dataset, which are conventional IID and non-IID as well

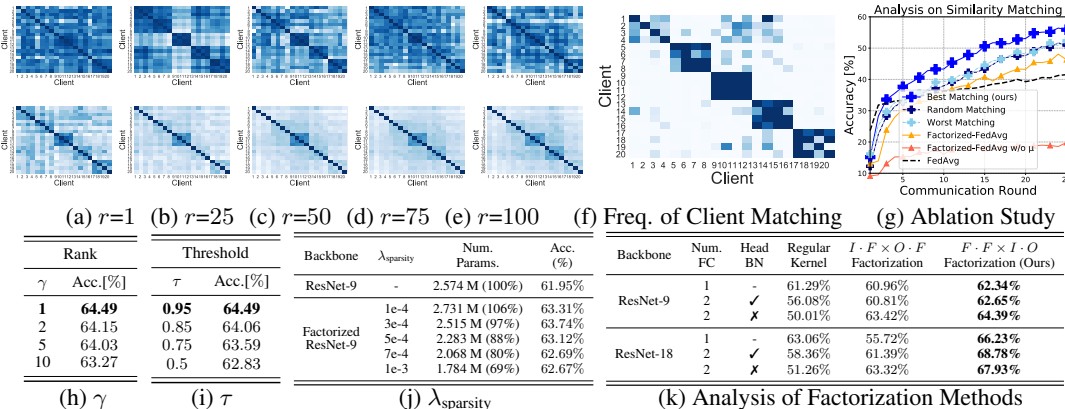

(a) $r=1$  (b) $r=25$  (c) $r=50$  (d) $r=75$  (e) $r=100$  (f) Freq. of Client Matching  (g) Ablation Study

| Rank | | Threshold | | Backbone | $\lambda_{sparsity}$ | Num. Params. | Acc. (%) | Backbone | Num. FC | Head BN | Regular Kernel | $I \cdot F \times O \cdot F$ Factorization | $F \cdot F \times I \cdot O$ Factorization (Ours) |
|---|---|---|---|---|---|---|---|---|---|---|---|---|
| $\gamma$ | Acc.[%] | $\tau$ | Acc.[%] | ResNet-9 | - | 2.574 M (100%) | 61.95% | ResNet-9 | 1 | - | 61.29% | 60.96% | **62.34%** |
| **1** | **64.49** | **0.95** | **64.49** | | 1e-4 | 2.731 M (106%) | 63.31% | | 2 | ✓ | 56.08% | 60.81% | **62.65%** |
| 2 | 64.15 | 0.85 | 64.06 | Factorized | 3e-4 | 2.515 M (97%) | 63.74% | | 2 | ✗ | 50.01% | 63.42% | **64.39%** |
| 5 | 64.03 | 0.75 | 63.59 | ResNet-9 | 5e-4 | 2.283 M (88%) | 63.12% | ResNet-18 | 1 | - | 63.06% | 55.72% | **66.23%** |
| 10 | 63.27 | 0.5 | 62.83 | | 7e-4 | 2.068 M (80%) | 62.69% | | 2 | ✓ | 58.36% | 61.39% | **68.78%** |
| | | | | | 1e-3 | 1.784 M (69%) | 62.67% | | 2 | ✗ | 51.26% | 63.32% | **67.93%** |

(h) $\gamma$      (i) $\tau$      (j) $\lambda_{sparsity}$      (k) Analysis of Factorization Methods

Figure 7: **Analysis on** `Factorized-FL` **algorithms (a)** and **(e)**: While $\mathbf{u}^{L-1}$ show strong similarity across all clients (upper row), $\mathbf{v}^{L-1}$ show exclusive similarity based on domains (bottom row). Darker colors indicate higher scores. **(f)**: the frequency of client matching after 100 rounds. **(g)**: component analysis. **(h)**, **(i)**, and **(j)**: analysis on $\gamma$, $\tau$, and $\lambda$. **(k)** the effect of how to factorize model parameters

as permuted IID and permuted non-IID, whose labels are permuted and incompatible to each other. We split the datasets into 20 partitions and then simply permuted the labels on the same partition for label permuted settings. (2) **Domain Heterogeneous Scenario**: we use CIFAR-100 datasets [11] and create five sub-datasets grouped by 10 similar classes, such as Household Objects, Fruits&Foods, Trees&Flowers, Transport, and Animals. We assign 4 clients for each sub-dataset, i.e. Client 1-4 to Household Objects, Client 5-8 to Fruits&Foods, Client 9-12 to Trees&Flowers, Client 13-16 to Transport, and Client 17-20 to Animals. We then permute the labels for all partitions to simulate further realistic scenarios. Further descriptions are elaborated in Appendix (Section B).

## 5.2 Experimental Result

**Label-heterogeneous FL** As shown in Table 1 (Top), for the standard IID and Non-IID settings, all FL methods obtain higher performance than the local training baseline (`Stand-Alone`), which confirms that the locally learned knowledge is beneficial to others, when the data and label distributions are homogeneous across clients. However, when the labels are not synchronized across all clients (Permuted IID/Non-IID), all previous FL methods suffer from significant performance degeneration, even lower than that of the local training baseline. Again, note that we do not share the classifier layers to ensure fairness across all algorithms in these permuted settings. We conjecture that this is caused by the label permutation leading the local model to evolve a permuted set of features that are not coordinate-wise compatible to others when aggregated. Contrarily, our method `Factorized-FL` $\alpha$ shows consistent performance regardless of whether the labels are permuted or not. `Factorized-FL` $\beta$ even largely outperforms all baselines with significantly superior performance. Test accuracy curves over the communication round and transmission cost are in Appendix (Section D).

**Domain-heterogeneous FL** Table 1 (Bottom) shows the experimental results for the domain and label heterogeneous scenarios. We observe that the conventional FL baselines, i.e. `FedAvg`, `FedProx`, fail to obtain better performance over purely local training baseline (`Stand-Alone`) due to the naive aggregation of extremely heterogeneous knowledge, which causes detrimental knowledge collapse. `FedFOMO` and `Clustered-FL` shows slightly higher performance (1-2%$p$) over `Stand-Alone` model, as they can avoid irrelevant clients when aggregating local knowledge. The other personalized FL methods, i.e. `Per-FedAvg` and `pFedPara`, also show 1-2%$p$ higher performance over `Stand-Alone` model as they are specialized for personalized FL scenarios. However, on average, our method largely outperforms all baseline models even with the smallest communication costs (`Factorized-FL` $\alpha$), as shown in Figure 6 (e). In the figures, we plot the convergence rate of our `Factorized-FL` $\alpha$ & $\beta$ over communication round (Top) and transmission cost (Bottom), compared to the baseline models. Both models consistently obtain superior performance compared to the other models. Particularly, unlike `pFedPara` which uses low rank matrices for knowledge sharing, `Factorized-FL` $\alpha$ only communicates with factorized vectors, such as $\mathcal{U}$ for base knowledge sharing and $\mathbf{v}_k^{L-1}$ for similarity matching and this allows our method to largely reduce the communication costs while showing the best performance as shown in Figure 6 (e) Bottom.

**Effect of kernel factorization** In Figure 7 (g), we perform an ablation study of our factorization method in the domain-heterogeneous scenario. To clearly see the effectiveness of our factorization methods, we compare `Factorized-FedAvg`, a variant of Factorized-FL $\beta$ without similarity matching, against `FedAvg`. As shown, `Factorized-FedAvg` achieves higher performance over the original `FedAvg` model. As the only difference between the two is whether kernel is factorized or not, this clearly demonstrates that our factorization method alone improves the model performance by alleviating knowledge collapse. We further analyze the effect of the sparse bias matrix $\mathcal{M}$. When we remove $\mathcal{M}$ from the `Factorized-FedAvg` model, `Factorized-FedAvg` w/o $\mu$ in the figure, we observe large performance drop. This shows that that the bias term is essential in compensating for the loss of expressiveness from the rather extreme factorization. With only $\mathcal{U}$ and $\mathcal{V}$, we use $90\%$ less model parameters ($0.27M$) compared to the regular kernel model ($2.574M$). Also, in Figure 7 (k), we observe that model factorized by $\mathbf{u} \in \mathbb{R}^{F \cdot F}$ and $\mathbf{v} \in \mathbb{R}^{I \cdot O}$ perform better than the model factorized by $\mathbf{u} \in \mathbb{R}^{I \cdot F}$ and $\mathbf{v} \in \mathbb{R}^{O \cdot F}$. This is because in the former case, the factorization will separate the base knowledge from the task-specific knowledge, as described in Figure 5 (Left), but in the latter case the factorization does not have such a natural interpretation. We also show the scalability and reliability of our factorization method with different architectures, consisting of convolutional, fully-connected layers, batch normalization, and skip connections as shown in Figure 7 (k).

**Effect of similarity matching** To verify the effectiveness of our similarity matching method using the personalized factorized vector $\mathbf{v}$, we visualize the inter-client similarity of $\mathbf{u}_{f_k}^{L-1}$ and $\mathbf{v}_{f_k}^{L-1}$ from the second last layer of 20 clients in a domain heterogeneous setting. As shown in Figure 7 from (a) round 1 to (e) round 100, we observe $\mathbf{u}_{f_k}^{L-1}$ (upper row) are indeed highly correlated with other clients (the darker color indicate higher similarity scores) while $\mathbf{v}_{f_k}^{L-1}$ (bottom row) are relatively uncorrelated to each other, as expected as our assumption that $\mathcal{U}_{f_k}$ capture base knowledge across all clients and $\mathcal{V}_{f_k}$ capture personalized knowledge. Further, we also observe that $\mathbf{v}_{f_k}^{L-1}$ obtains higher similarity to that of the parameters trained on the same domains (but with permuted labels), i.e. Client 1-4, Client 5-8, Client 9-12, Client 13-16, and Client 17-20, showing that they are effective in capturing task- and domain-level similarities across models. We further visualize the frequency of the client matching across clients in Figure (f), after 100 communication rounds. With only a single vector parameter $\mathbf{v}_{f_k}^{L-1}$, our method finds which clients's knowledge will be helpful to which clients. For further analysis on our similarity matching, we compare it against Random and Worst Matching baselines under the multi domain scenario. The random matching baseline randomly selects three arbitrary models to be aggregated at each round, while the worst matching baseline selects three most dissimilar models. As shown in Table 7 (g), both baselines significantly suffer from the performance degeneration compared to our similarity matching algorithm, which shows its effectiveness.

**Analysis on $\gamma$, $\tau$, and $\lambda_{\mathbf{sparsity}}$** Figure 7 (h) and (i) show the analysis of the hyperparameters, $\gamma$ and $\tau$, on the domain-heterogeneous dataset. As shown in Figure 7 (h), the performance slightly decreases with larger $\gamma$, since higher rank weight matrices may result in larger coordinate-wise incompatibility of the parameters, which could lead to knowledge collapse when performing aggregation. Also, the lower $\tau$s show the lower performance, as less similar knowledge can be aggregated more frequently. In Figure 7 (j), we train Client 1 on CIFAR-10 IID Partition 1 for 20 epochs using both factorized and regular models. As shown, our factorized model outperforms regular models ($2.574M$) even with $30\%$ less parameters ($1.784M$). We further analyze the sparsity in Section D.2 of Appendix.

## 6 Conclusion

We introduced a realistic federated learning scenario where the labeling schemes are not synchronized across all participants (label heterogeneity) and the tasks and the domains tackled by each local model are different from those of others (domain heterogeneity). We then proposed a novel method to tackle this problem, whose local model weights are factorized into the product of two vectors plus a sparse bias term. We then aggregate only the first vectors, for them to capture the common knowledge across clients, while allowing the other vectors and the sparse bias term to be client-specific, accounting for label and domain heterogeneity. Further, we use the client-specific vectors to measure the similarity scores across local models, which are then used for weighted averaging, for personalized federated learning of each local model. Our method not only avoids knowledge collapse from aggregating incompatible parameters across heterogeneous models, but also significantly reduces the communication costs. We validate our method on both label and domain heterogeneous settings, on which it largely outperforms relevant baselines.

## Limitation and Potential Societal Impacts

**Limitations** Federated Learning (FL) is devised to protect the **privacy of sensitive personal data** by strictly avoiding sharing them with other users, institutions, and companies. The common limitation in FL is that, even though almost all existing methods regard sharing locally learned parameters of gradients as safe, they still may contain some possibility to reproduce the local data distribution, which violates the preservation of privacy. Although our work alleviates this probability by significantly reducing the dimensionality of parameters and partially sharing them, it may be still sub-optimal in an aspect of security, which is an open research domain for federated learning.

**Potential Societal Impact** Federated learning is a collaborative learning framework where multiple clients can actively participate in the learning procedure anywhere and anytime. Nowadays, numerous terminal devices, i.e. smartphones, smartwatches, IoT devices, etc, are utilized as participants of the collaborative learning scheme. However, as the number of participants in federated learning increases, the number of communications between them increases accordingly. Such situations leading to the exhaustive **communication costs** may worsen **the global environmental crisis**, i.e. global warming. Even though our method can alleviate such risks, since ours significantly reduces the communication costs by partially sharing the factorized vectors, it is essential to develop further optimal solutions for compressing the knowledge in both efficient and effective ways for federated learning that the massive number of participants collaborate.

## Acknowledgments and Disclosure of Funding

This work was supported by Institute of Information & communications Technology Planning & Evaluation (IITP) grant funded by the Korea government(MSIT) (No.2019-0-00075, Artificial Intelligence Graduate School Program(KAIST)). It was also results of a study on the "HPC Support" Project, supported by the 'Ministry of Science and ICT' and NIPA.

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
