**Organization** We provide in-depth descriptions for our algorithms, experimental setups, i.e. dataset configurations, implementation & training details, and additional experimental results & analysis that are not covered in the main document, as organized as follows:

- **Section A**: We provide the pseudo-code algorithms for `Factorized-FL`.

- **Section B**: We describe dataset configurations for label- and domain-heterogenous scenarios.

- **Section C**: We elaborate on implementation and training details for our methods and the baselines.

- **Section D**: We provide additional experimental results and analysis.

# A `Factorized-FL` Algorithms

In this section, we describe our pseudo-code algorithms for `Factorized-FL` $\alpha$ and `Factorized-FL` $\beta$ in Algorithm 1 and 2. Our `Factorized-FL` $\alpha$ has strength for not only reducing the dimensionality of model parameters by factorizing them into rank 1 vector spaces and the additional highly-sparse matrices, but also effectively learning client-general and task-specific knowledge, performing well on domain heterogeneous scenario. Particularly, `Factorized-FL` $\alpha$ transmits a small portion of the models which are a set of $\mathbf{u}$ ($\mathcal{U}$) and a single vector $\mathbf{v}^{L-1}$ form the second last layer of neural networks, which significantly reduces communication costs while showing strong performance in label- and domain-heterogeneous scenarios, as shown in Section 5 in the main document.

---

**Algorithm 1** `Factorized-FL` $\alpha$

1: $R$: number of rounds, $E$: number of epochs, $K$: number of clients, $\mathcal{F}$: a set of clients, $\Omega(\cdot)$: similarity matching function, $\sigma$: client-wise similarity score, $L$: number of layers in neural networks. $\mathcal{U}_k$, $\mathcal{V}_k$, and $\mathcal{M}_k$: factorized parameters.
2: **Function** RunServer()
3: initialize $\mathcal{F}$
4: **for** each round $r = 1, 2, \ldots, R$ **do**
5:    $\mathcal{F}^{(r)} \leftarrow$ select $K^{(r)}$ clients from $\mathcal{F}$
6:    **for** each client $f_k^{(r)} \in \mathcal{F}^{(r)}$ **in parallel do**
7:      **if** $r > 1$ **then**
8:       $\{\sigma_i\}_{i \neq k}^{K^{(r)}} \leftarrow \Omega(\mathbf{v}_{f_k}^{L-1}, \mathbf{v}_{f_{i \neq k:K^{(r)}}}^{L-1})$
9:       $\mathcal{U}_k^{(r)} \leftarrow \frac{\exp(\epsilon \cdot \sigma_i)}{\sum_{i=1}^{K^{(r)}} \exp(\epsilon \cdot \sigma_i)} \sum_{i=1}^{K^{(r)}} \mathcal{U}_i^{(r)}$
10:      **end if**
11:      $\mathcal{U}_k^{(r+1)}, \mathbf{v}_{f_k}^{L-1} \leftarrow$ RunClient$(\mathcal{U}_k^{(r)})$
12:    **end for**
13: **end for**
14: **Function** RunClient$(\mathcal{U}_k)$
15: $\theta_k \leftarrow \mathcal{U}_k \times \mathcal{V}_k \oplus \mathcal{M}_k$
16: **for** each local epoch $e$ from 1 to $E$ **do**
17:    **for** minibatch $\mathcal{B} \in \mathcal{D}_k$ **do**
18:      $\theta_k \leftarrow \theta_k - \eta \nabla \mathcal{L}(\mathcal{B}; \theta_k)$
19:    **end for**
20: **end for**
21: **return** $\mathcal{U}_k, \mathbf{v}_{f_k}^{L-1}$

---

**Algorithm 2** `Factorized-FL` $\beta$

1: **Function** RunServer()
2: initialize $\mathcal{F}$
3: **for** each round $r = 1, 2, \ldots, R$ **do**
4:    $\mathcal{F}^{(r)} \leftarrow$ select $K^{(r)}$ clients from $\mathcal{F}$
5:    **for** each client $f_k^{(r)} \in \mathcal{F}^{(r)}$ **in parallel do**
6:      **if** $r > 1$ **then**
7:       $\{\sigma_i\}_{i \neq k}^{K^{(r)}} \leftarrow \Omega(\mathbf{v}_{f_k}^{L-1}, \mathbf{v}_{f_{i \neq k:K^{(r)}}}^{L-1})$
8:       $\mathcal{U}_k^{(r)} \leftarrow \frac{\exp(\epsilon \cdot \sigma_i)}{\sum_{i=1}^{K^{(r)}} \exp(\epsilon \cdot \sigma_i)} \sum_{i=1}^{K^{(r)}} \mathcal{U}_i^{(r)}$
9:       $\mathcal{V}_k^{(r)} \leftarrow \frac{\exp(\epsilon \cdot \sigma_i)}{\sum_{i=1}^{K^{(r)}} \exp(\epsilon \cdot \sigma_i)} \sum_{i=1}^{K^{(r)}} \mathcal{V}_i^{(r)}$
10:       $\mathcal{M}_k^{(r)} \leftarrow \frac{\exp(\epsilon \cdot \sigma_i)}{\sum_{i=1}^{K^{(r)}} \exp(\epsilon \cdot \sigma_i)} \sum_{i=1}^{K^{(r)}} \mathcal{M}_i^{(r)}$
11:      **end if**
12:      $\mathcal{U}_k^{(r+1)}, \mathcal{V}_k^{(r+1)}, \mathcal{M}_k^{(r+1)}$
13:        $\leftarrow$ RunClient$(\mathcal{U}_k^{(r)}, \mathcal{V}_k^{(r)}, \mathcal{M}_k^{(r)})$
14:    **end for**
15: **end for**
16: **Function** RunClient$(\mathcal{U}_k, \mathcal{V}_k, \mathcal{M}_k)$)
17: $\theta_k \leftarrow \mathcal{U}_k \times \mathcal{V}_k \oplus \mathcal{M}_k$
18: **for** each local epoch $e$ from 1 to $E$ **do**
19:    **for** minibatch $\mathcal{B} \in \mathcal{D}_k$ **do**
20:      $\theta_k \leftarrow \theta_k - \eta \nabla \mathcal{L}(\mathcal{B}; \theta_k)$
21:    **end for**
22: **end for**
23: **return** $\mathcal{U}_k, \mathcal{V}_k, \mathcal{M}_k$

---

# B Dataset Configurations

In this section, we describe detailed configurations for datasets that we used in label- and domain-heterogeneous scenarios.

Table 2: **Label permutations for label-heterogeneous scenario** We provide permutations of labels for each dataset. These permutations are randomly generated based on different seeds.

| Dataset | Class | Original Labels | Client No. | | | | | | | | | | | | | | | | | | | |
|---|---|---|---|---|---|---|---|---|---|---|---|---|---|---|---|---|---|---|---|---|---|---|---|
| | | | 1 | 2 | 3 | 4 | 5 | 6 | 7 | 8 | 9 | 10 | 11 | 12 | 13 | 14 | 15 | 16 | 17 | 18 | 19 | 20 |
| CIFAR-10 | Airplane | 0 | 2 | 5 | 3 | 0 | 8 | 2 | 4 | 4 | 2 | 2 | 0 | 6 | 1 | 8 | 4 | 0 | 0 | 6 | 9 | 7 |
| | Automobile | 1 | 8 | 4 | 1 | 5 | 1 | 8 | 1 | 0 | 9 | 7 | 7 | 3 | 6 | 2 | 0 | 4 | 1 | 8 | 1 | 4 |
| | Bird | 2 | 3 | 0 | 5 | 3 | 4 | 9 | 9 | 5 | 4 | 1 | 5 | 5 | 3 | 0 | 1 | 6 | 3 | 7 | 6 | 5 |
| | Cat | 3 | 5 | 9 | 0 | 7 | 9 | 3 | 8 | 8 | 7 | 6 | 2 | 7 | 8 | 7 | 6 | 7 | 4 | 4 | 3 | 8 |
| | Deer | 4 | 6 | 2 | 6 | 9 | 6 | 5 | 2 | 1 | 0 | 9 | 6 | 1 | 7 | 3 | 7 | 5 | 5 | 3 | 8 | 9 |
| | Dog | 5 | 4 | 1 | 4 | 8 | 5 | 6 | 6 | 6 | 3 | 4 | 9 | 4 | 4 | 4 | 5 | 9 | 8 | 1 | 2 | 0 |
| | Frog | 6 | 9 | 3 | 2 | 1 | 2 | 0 | 3 | 2 | 6 | 3 | 3 | 0 | 5 | 6 | 2 | 1 | 7 | 2 | 5 | 1 |
| | Horse | 7 | 0 | 7 | 9 | 4 | 3 | 7 | 0 | 3 | 5 | 0 | 1 | 2 | 9 | 5 | 3 | 2 | 9 | 5 | 0 | 6 |
| | Ship | 8 | 1 | 8 | 7 | 6 | 7 | 4 | 5 | 7 | 8 | 5 | 8 | 9 | 0 | 9 | 8 | 3 | 6 | 0 | 7 | 2 |
| | Truck | 9 | 7 | 6 | 8 | 2 | 0 | 1 | 7 | 9 | 1 | 8 | 4 | 8 | 2 | 1 | 9 | 8 | 2 | 9 | 4 | 3 |
| SVHN | Digit 0 | 10 | 2 | 5 | 3 | 0 | 8 | 2 | 4 | 4 | 2 | 2 | 0 | 6 | 1 | 8 | 4 | 0 | 0 | 6 | 9 | 7 |
| | Digit 1 | 1 | 8 | 4 | 1 | 5 | 1 | 8 | 1 | 0 | 9 | 7 | 7 | 3 | 6 | 2 | 0 | 4 | 1 | 8 | 1 | 4 |
| | Digit 2 | 2 | 3 | 0 | 5 | 3 | 4 | 9 | 9 | 5 | 4 | 1 | 5 | 5 | 3 | 0 | 1 | 6 | 3 | 7 | 6 | 5 |
| | Digit 3 | 3 | 5 | 9 | 0 | 7 | 9 | 3 | 8 | 8 | 7 | 6 | 2 | 7 | 8 | 7 | 6 | 7 | 4 | 4 | 3 | 8 |
| | Digit 4 | 4 | 6 | 2 | 6 | 9 | 6 | 5 | 2 | 1 | 0 | 9 | 6 | 1 | 7 | 3 | 7 | 5 | 5 | 3 | 8 | 9 |
| | Digit 5 | 5 | 4 | 1 | 4 | 8 | 5 | 6 | 6 | 6 | 3 | 4 | 9 | 4 | 4 | 4 | 5 | 9 | 8 | 1 | 2 | 0 |
| | Digit 6 | 6 | 9 | 3 | 2 | 1 | 2 | 0 | 3 | 2 | 6 | 3 | 3 | 0 | 5 | 6 | 2 | 1 | 7 | 2 | 5 | 1 |
| | Digit 7 | 7 | 0 | 7 | 9 | 4 | 3 | 7 | 0 | 3 | 5 | 0 | 1 | 2 | 9 | 5 | 3 | 2 | 9 | 5 | 0 | 6 |
| | Digit 8 | 8 | 1 | 8 | 7 | 6 | 7 | 4 | 5 | 7 | 8 | 5 | 8 | 9 | 0 | 9 | 8 | 3 | 6 | 0 | 7 | 2 |
| | Digit 9 | 9 | 7 | 6 | 8 | 2 | 0 | 1 | 7 | 9 | 1 | 8 | 4 | 8 | 2 | 1 | 9 | 8 | 2 | 9 | 4 | 3 |

## B.1 Label Heterogeneous Scenario

We use CIFAR-10 and SVHN for the label-heterogeneous scenario. We first split each dataset into train, validation, and test sets for CIFAR-10 ($48,000/6,000/6,000$) and SVHN ($79,431/9,929/9,929$). We then split the train set into $K$ local partitions $\mathcal{P}_{1:20}$ ($K$=20) for IID partitions (all instances in each class are evenly distributed to all clients) or for the non-IID partitions (instances in each class are sampled from Dirichlet distribution with $\alpha$=0.5). We further permute the labels for each class per local partition $\mathcal{P}_k$ for permuted IID and permuted non-IID scenarios. We use different random seed per client, i.e. `fixed global seed + client id`, for example, $1234 + 0$ for Client 1 and $1234 + 19$ for Client 20. We provide permutations of labels that we used for each dataset in Table 2.

## B.2 Domain Heterogeneous Scenario

We use CIFAR-100 datasets ($60,000$) and create five sub-datasets grouped by 10 similar classes, such as Fruits&Foods ($6,000$), Transport ($6,000$), Household Objects ($6,000$), Animals ($6,000$), Trees&Flowers ($6,000$). We then split train ($4,800$), test ($600$), validation ($600$) sets for each sub-datset. To have 20 clients in total, we assign four clients per subdataset, and split each train set into $4$ partitions, making a single partition contains $1,200$ instances. Additionally, we further permute the labels for those 20 partitions to simulate more realistic scenarios where labeling schemes are not synchronized across all clients even in the same domain (sub-dataset). We provide class division and label permutation information in Table 3.

# C Training Details & Implementations

In this section, we provide detailed implementation and training details that are not described in the main document.

## C.1 Baseline Models

`Stand-Alone` does not share their locally learned knowledge with other clients. It shows the pure model performance on the data partitions. In our extremely heterogeneous scenarios, where knowledge collapse may happen severely, which even deteriorates the local knowledge, this model shows the comparable performance amongst the existing models.

`FedAvg` [18] performs weighted aggregation of the model parameters, considering the size of the local training set. This model is considered as the standard baseline of many federated learning algorithms.

Table 3: **Class division and label permutation information for domain-heterogeneous scenario** We provide class division information and label permutation details for each domain. These permutations are randomly generated based on the same method used in label-heterogeneous scenario using different seeds.

| Domain | Class | Original Labels | 1 | 2 | 3 | 4 | 5 | 6 | 7 | 8 | 9 | 10 | 11 | 12 | 13 | 14 | 15 | 16 | 17 | 18 | 19 | 20 |
|---|---|---|---|---|---|---|---|---|---|---|---|---|---|---|---|---|---|---|---|---|---|---|
| Household Objects | Bed | 5 | 2 | 5 | 3 | 0 | - | - | - | - | - | - | - | - | - | - | - | - | - | - | - | - |
| | Chair | 20 | 8 | 4 | 1 | 5 | - | - | - | - | - | - | - | - | - | - | - | - | - | - | - | - |
| | Couch | 22 | 3 | 0 | 5 | 3 | - | - | - | - | - | - | - | - | - | - | - | - | - | - | - | - |
| | Table | 25 | 5 | 9 | 0 | 7 | - | - | - | - | - | - | - | - | - | - | - | - | - | - | - | - |
| | Wardrobe | 39 | 6 | 2 | 6 | 9 | - | - | - | - | - | - | - | - | - | - | - | - | - | - | - | - |
| | Clock | 40 | 4 | 1 | 4 | 8 | - | - | - | - | - | - | - | - | - | - | - | - | - | - | - | - |
| | Keyboard | 84 | 9 | 3 | 2 | 1 | - | - | - | - | - | - | - | - | - | - | - | - | - | - | - | - |
| | Lamp | 86 | 0 | 7 | 9 | 4 | - | - | - | - | - | - | - | - | - | - | - | - | - | - | - | - |
| | Telephone | 87 | 1 | 8 | 7 | 6 | - | - | - | - | - | - | - | - | - | - | - | - | - | - | - | - |
| | Television | 94 | 7 | 6 | 8 | 2 | - | - | - | - | - | - | - | - | - | - | - | - | - | - | - | - |
| Fruits & Foods | Apple | 0 | - | - | - | - | 8 | 2 | 4 | 4 | - | - | - | - | - | - | - | - | - | - | - | - |
| | Mushroom | 9 | - | - | - | - | 1 | 8 | 1 | 0 | - | - | - | - | - | - | - | - | - | - | - | - |
| | Orange | 10 | - | - | - | - | 4 | 9 | 9 | 5 | - | - | - | - | - | - | - | - | - | - | - | - |
| | Pear | 16 | - | - | - | - | 9 | 3 | 8 | 8 | - | - | - | - | - | - | - | - | - | - | - | - |
| | Sweet Pepper | 28 | - | - | - | - | 6 | 5 | 2 | 1 | - | - | - | - | - | - | - | - | - | - | - | - |
| | Bottle | 51 | - | - | - | - | 5 | 6 | 6 | 6 | - | - | - | - | - | - | - | - | - | - | - | - |
| | Bowl | 53 | - | - | - | - | 2 | 0 | 3 | 2 | - | - | - | - | - | - | - | - | - | - | - | - |
| | Can | 57 | - | - | - | - | 3 | 7 | 0 | 3 | - | - | - | - | - | - | - | - | - | - | - | - |
| | Cup | 61 | - | - | - | - | 7 | 4 | 5 | 7 | - | - | - | - | - | - | - | - | - | - | - | - |
| | Plate | 83 | - | - | - | - | 0 | 1 | 7 | 9 | - | - | - | - | - | - | - | - | - | - | - | - |
| Trees & Flowers | Orchid | 47 | - | - | - | - | - | - | - | - | 2 | 2 | 0 | 6 | - | - | - | - | - | - | - | - |
| | Poppy | 52 | - | - | - | - | - | - | - | - | 9 | 7 | 7 | 3 | - | - | - | - | - | - | - | - |
| | Rose | 54 | - | - | - | - | - | - | - | - | 4 | 1 | 5 | 5 | - | - | - | - | - | - | - | - |
| | Sunflower | 56 | - | - | - | - | - | - | - | - | 7 | 6 | 2 | 7 | - | - | - | - | - | - | - | - |
| | Tulip | 59 | - | - | - | - | - | - | - | - | 0 | 9 | 6 | 1 | - | - | - | - | - | - | - | - |
| | Maple Tree | 62 | - | - | - | - | - | - | - | - | 3 | 4 | 9 | 4 | - | - | - | - | - | - | - | - |
| | Oak Tree | 70 | - | - | - | - | - | - | - | - | 6 | 3 | 3 | 0 | - | - | - | - | - | - | - | - |
| | Palm Tree | 82 | - | - | - | - | - | - | - | - | 5 | 0 | 1 | 2 | - | - | - | - | - | - | - | - |
| | Pine Tree | 92 | - | - | - | - | - | - | - | - | 8 | 5 | 8 | 9 | - | - | - | - | - | - | - | - |
| | Willow Tree | 96 | - | - | - | - | - | - | - | - | 1 | 8 | 4 | 8 | - | - | - | - | - | - | - | - |
| Transport | Lawn Mower | 8 | - | - | - | - | - | - | - | - | - | - | - | - | 1 | 8 | 4 | 0 | - | - | - | - |
| | Rocket | 13 | - | - | - | - | - | - | - | - | - | - | - | - | 6 | 2 | 0 | 4 | - | - | - | - |
| | Streetcar | 41 | - | - | - | - | - | - | - | - | - | - | - | - | 3 | 0 | 1 | 6 | - | - | - | - |
| | Tank | 48 | - | - | - | - | - | - | - | - | - | - | - | - | 8 | 7 | 6 | 7 | - | - | - | - |
| | Tractor | 58 | - | - | - | - | - | - | - | - | - | - | - | - | 7 | 3 | 7 | 5 | - | - | - | - |
| | Bicycle | 69 | - | - | - | - | - | - | - | - | - | - | - | - | 4 | 4 | 5 | 9 | - | - | - | - |
| | Bus | 81 | - | - | - | - | - | - | - | - | - | - | - | - | 5 | 6 | 2 | 1 | - | - | - | - |
| | Motorcycle | 85 | - | - | - | - | - | - | - | - | - | - | - | - | 9 | 5 | 3 | 2 | - | - | - | - |
| | Pickup Truck | 89 | - | - | - | - | - | - | - | - | - | - | - | - | 0 | 9 | 8 | 3 | - | - | - | - |
| | Train | 90 | - | - | - | - | - | - | - | - | - | - | - | - | 2 | 1 | 9 | 8 | - | - | - | - |
| Animals | Fox | 3 | - | - | - | - | - | - | - | - | - | - | - | - | - | - | - | - | 0 | 6 | 9 | 7 |
| | Porcupine | 34 | - | - | - | - | - | - | - | - | - | - | - | - | - | - | - | - | 1 | 8 | 1 | 4 |
| | Possum | 42 | - | - | - | - | - | - | - | - | - | - | - | - | - | - | - | - | 3 | 7 | 6 | 5 |
| | Raccoon | 43 | - | - | - | - | - | - | - | - | - | - | - | - | - | - | - | - | 4 | 4 | 3 | 8 |
| | Skunk | 63 | - | - | - | - | - | - | - | - | - | - | - | - | - | - | - | - | 5 | 3 | 8 | 9 |
| | Bear | 64 | - | - | - | - | - | - | - | - | - | - | - | - | - | - | - | - | 8 | 1 | 2 | 0 |
| | Leopard | 66 | - | - | - | - | - | - | - | - | - | - | - | - | - | - | - | - | 7 | 2 | 5 | 1 |
| | Lion | 75 | - | - | - | - | - | - | - | - | - | - | - | - | - | - | - | - | 9 | 5 | 0 | 6 |
| | Tiger | 88 | - | - | - | - | - | - | - | - | - | - | - | - | - | - | - | - | 6 | 0 | 7 | 2 |
| | Wolf | 97 | - | - | - | - | - | - | - | - | - | - | - | - | - | - | - | - | 2 | 9 | 4 | 3 |

`FedProx` [15]   uses proximal regularization term for alleviating divergence between global parameters and the local parameters. This model is devised to tackle data heterogeneity (also known as the non-iid problems) across the clients.

`Clustered-FL` [23]   performs bi-partitioning process for the participants when certain conditions are satisfied (calculating the norm of the gradient updates and comparing it with given threshold values). It continues to split the given pool of the clients into two novel clusters and performs knowledge aggregations for each cluster.

`pFedPara` [20]   uses the Hadamard product of two sets of low rank matrices and reconstructs the kernel parameters. It shares one of the sets of low rank matrices with other clients and remains another set of low rank matrices for the personalization.

`Per-FedAvg` [5]    adopts Model-Agnostic Meta Learning (MAML) [6] approach to federated learning algorithm to search the initial global model which participants can easily adapt to their local data by being trained with only one or a few steps of gradient descent.

`FedFOMO` [28]    leverages other clients' knowledge for improving their local models. It downloads a few random models from other clients and validates them on their own validation set at each client. When some parameters show better validation performance compared to their own local performance, it aggregates such helpful parameters with their own local parameters.

## C.2    Training Details

As default, all training configurations are equally set across all models, unless otherwise stated to ensure stricter fairness. We use ResNet-9 architecture as local backbone networks and train them on $32 \times 32$ sized images with 256 for batch size. We apply data augmentations, i.e. cropping, flipping, jittering, etc, during training. Optimizer that we used is Stochastic Gradient Descent (SGD). We set 1e-3 for learning rate, 1e-6 for weight decay, and 0.9 for momentum. For baseline models, we use the reported hyper-parameters as default, or we adjust hyper-parameters so that they show the best performance for fairness. For ours and `pFedPara`, the model capacity is adjusted to around 90% - 99% of the original size, as we fairly compare with other methods that use full capacity ($2.57M$ number of parameters). For ours, we use [5e-4, 1e-3] for $\lambda_{\text{sparsity}}$, [0-0.75] for $\tau$, [1, 20] for $\epsilon$. We use 8 GPUs (NVIDIA Titan Xp) for experiments.

## C.3    ResNet-9 Architecture

We use ResNet-9 architecture consisting of eight convolutional layers and one fully connected layer as a classifier, as described in Table 4. We use max pooling with size 2 after Conv 5 and an adaptive max pooling after Conv 8 to make output width 1 for the following FC layer. The total number of parameters of the model is $2.57M$. As we use

Table 4: **Detailed ResNet-9 Architecture**

| Layer | Input | Output | Filter Size | Stride | Dimension of $\mathbf{W}^l$ |
|-------|-------|--------|-------------|--------|------------------------------|
| Conv 1 | 3 | 64 | 3 | 1 | $64 \times 3 \times 3 \times 3$ |
| Conv 2 | 64 | 128 | 5 | 2 | $128 \times 64 \times 5 \times 5$ |
| Conv 3 | 128 | 128 | 3 | 1 | $128 \times 128 \times 3 \times 3$ |
| Conv 4 | 128 | 128 | 3 | 1 | $128 \times 128 \times 3 \times 3$ |
| Conv 5 | 128 | 256 | 3 | 1 | $256 \times 128 \times 3 \times 3$ |
| Conv 6 | 256 | 256 | 3 | 1 | $256 \times 256 \times 3 \times 3$ |
| Conv 7 | 256 | 256 | 3 | 1 | $256 \times 256 \times 3 \times 3$ |
| Conv 8 | 256 | 256 | 3 | 1 | $256 \times 256 \times 3 \times 3$ |
| FC 1 | 256 | $C$ | - | - | $256 \times C$ |

`PyTorch` framework for implementation and the default data type of tensor of the framework is 32-bits floating point, the model size can be calculated as $2.57 \times 4 = 10.28$ Mbytes.

## C.4    Calculation of Communication Cost

We measure the communication cost by $\{(P_{S2C} + P_{C2S}) \times 4\}_{byte} \times K \times R$, where $P_{S2C}$ is number of server-to-client transmitted parameters and $P_{C2S}$ is number of client-to-server transmitted parameters. Depending on the FL algorithms, $P_{S2C}$ and $P_{C2S}$ are differently calculated. For example, `FedFOMO` downloads a few random models from the server (10 by default as reported in the paper) but sends only a single local model to the server. Our `Factorized-FL` $\alpha$ only sends the small portion of model parameters, $\mathcal{U}$ and $\mathbf{v}^{L-1}$, to server, while receiving a single set of $\mathcal{U}$ from server.

# D    Additional Experimental Results

Table 5: **The Approximated Space-Time Analysis** We provide the approximated space-time analysis for training, inference, client-to-server communication costs, and server-to-client communication costs.

| Method | Training | Inference | C2S Cost | S2C Cost |
|--------|----------|-----------|----------|----------|
| FedAvg | $W \cdot N$ | $W \cdot N$ | $W$ | $W$ |
| pFedPara | $(U + V)^2 \cdot N$ | $(U + V)^2 \cdot N$ | $U + V$ | $U + V$ |
| FedFOMO | $W \cdot N + H \cdot W \cdot N_{val}$ | $W \cdot N$ | $W$ | $(H + 1) \cdot W$ |
| Factorized-FL $\alpha$ | $(U \cdot V + M) \cdot N$ | $(U \cdot V + M) \cdot N$ | $U + \mathbf{v}$ | $U$ |
| Factorized-FL $\beta$ | $(U \cdot V + M) \cdot N$ | $(U \cdot V + M) \cdot N$ | $U + V + M$ | $U + V + M$ |

### D.1 The Space-Time Analysis

We provide the approximated space-time complexity in Table 5, compared against those of the essential baselines. Let $W$ be the model size, $N$ be the number of instances, $H$ be the number of other models (for `FedFOMO` algorithms), $U$ be the size of the rank-1 vectors (or matrices for `pFedPara`) in $\mathcal{U}$, $V$ be that of $\mathcal{V}$, $M$ be the that of $\mathcal{M}$, $\mathbf{v}$ be the vector for the second last layer from $\mathcal{V}$. For training procedure of `FedFOMO`, it requires validation step for $H$ number of other model weights on their validation set at each local client. Note that, for communication costs of our methods (`Fac.-FL`), $(U \cdot V + M) \approx W$ and thus $(U + V + M) < W$ for both Client-to-Server (C2S) and Server-to-Client (S2C) communication costs. Please also refer to the actual size of the data transmission for the communication cost in Table 1.

### D.2 Sparsity Analysis on FL Scenarios

In the main document, we show the effect of model size and sparsity controlled by $\lambda_{\text{sparsity}}$ for a single model. In this section, we analyze it under federated learning scenario. In Figure 8 (a), we show the performance over model size in domain heterogeneous scenarios. As shown, our method shows superior performance even with around $65\%$ of the model size over the baseline model that achieves the best performance (`Per-FedAvg`) amongst other baseline models. With $50\%$ sparsity, ours still shows competitive performance compared to `Clustered-FL` and `FedAvg`, while it starts being significantly degenerated when sparsity becomes over $50\%$.

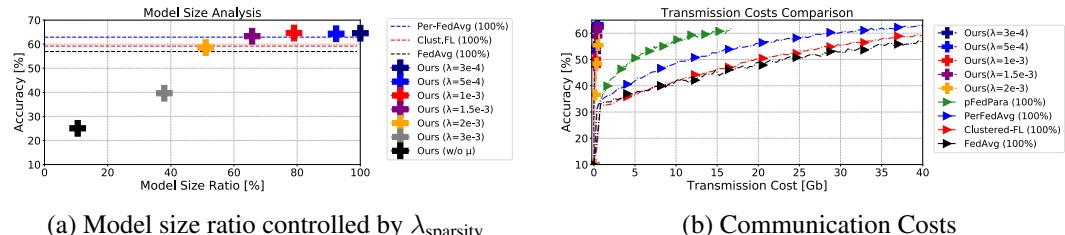

(a) Model size ratio controlled by $\lambda_{\text{sparsity}}$         (b) Communication Costs

Figure 8: **Model size and communication costs comparison** (a) we plot accuracy over model size on domain heterogeneous scenario. (b) we plot accuracy over transmission costs on domain heterogeneous scenario.

In Figure 8 (b), we show accuracy over communication costs. Note that, in our method, the model size is not really related to the communication costs since we send very small portion of model parameters. For example, even though we use almost full model size ($\lambda_{\text{sparsity}}$=3$e$-4), our communication cost is significantly lesser than the other baseline models, as shown in the figure.

### D.3 Test Accuracy Curves and Plots for Communication Costs

For label-heterogeneous FL scenario (Table 1 (Top)), we provide all test accuracy curves over communication rounds and transmission costs for results of CIFAR-10 and SVHN with stardard IID/non-IID and permuted IID/non-IID partitions in Figure 10. For domain-heterogeneous FL scenario (Table 1 (Bottom)), we provide performance of 20 clients In Figure 11.

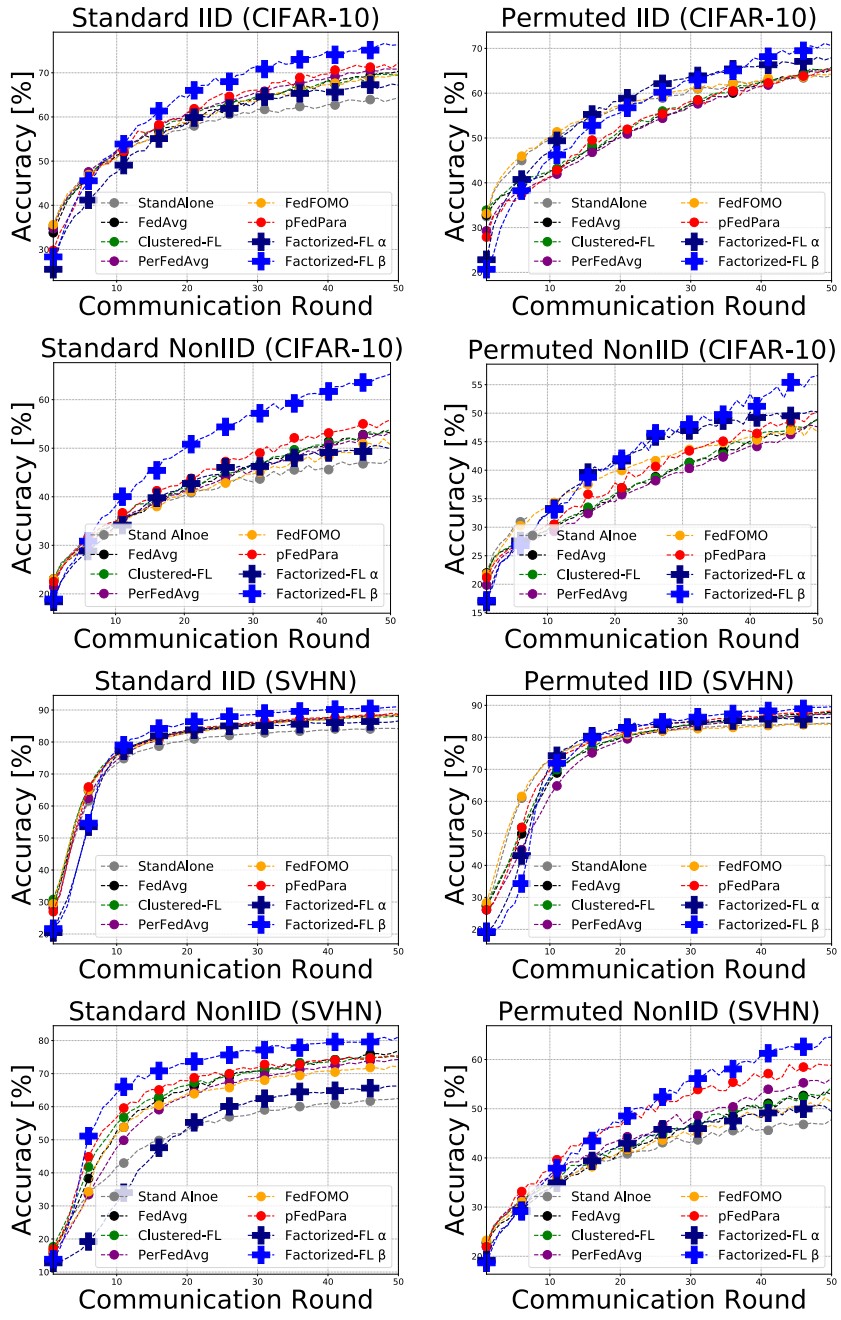

Figure 9: **Test accuracy curves over communication round for standard federated learning and label-heterogeneous FL scenario**: We provide test accuracy curves on CIFAR-10 and SVHN in standard iid/non-iid and permuted iid/non-iid partitions ($E$=5,$R$=50).

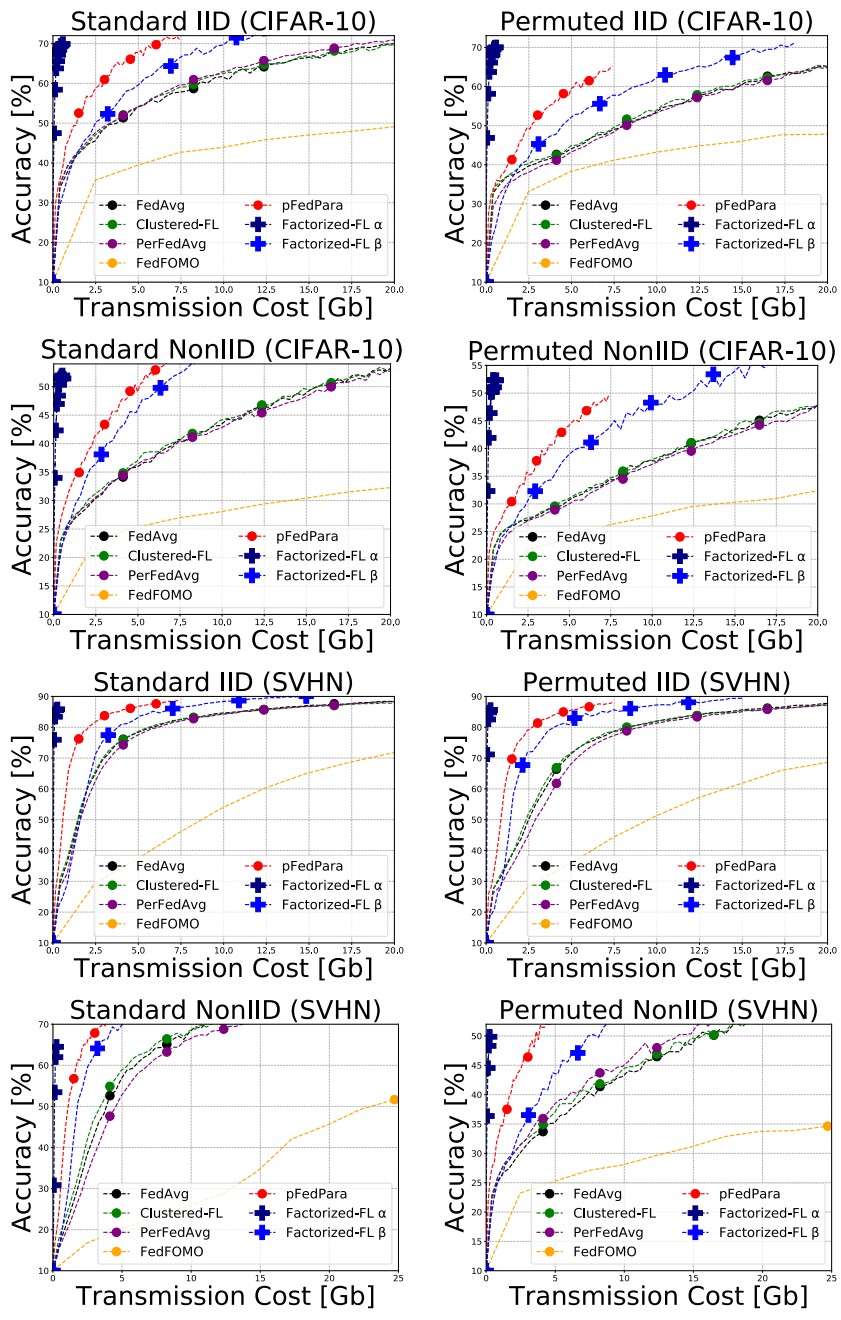

Figure 10: **Test accuracy over communication costs for standard federated learning and label-heterogeneous FL scenario**: We provide test accuracy curves on CIFAR-10 and SVHN in standard iid/non-iid and permuted iid/non-iid partitions ($E$=5,$R$=50).

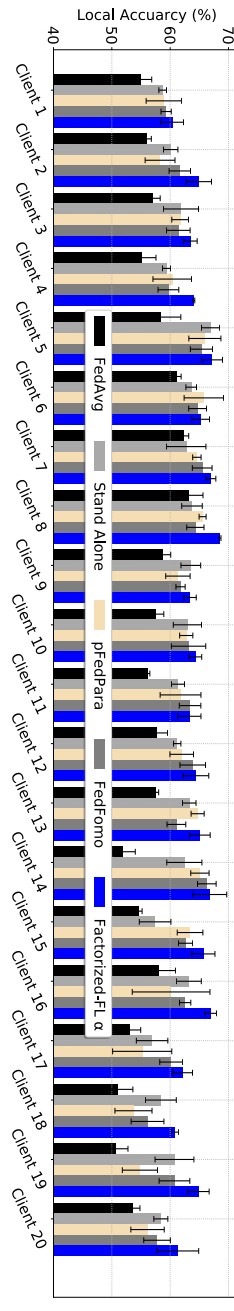

Figure 11: **Performance of all** 20 **clients in domain heterogeneous scenario**: We plot performance of 20 clients in domain-heterogeneous scenario, of which results are corresponding to Table 1 (Bottom).