# OpenReview forum: "Factorized-FL: Personalized Federated Learning with Parameter Factorization & Similarity Matching"
_NeurIPS.cc/2022/Conference — NeurIPS 2022 Accept_

### Official Review · Reviewer_dsaH · 2022-07-05

**Rating:** 4
**Confidence:** 4
**Soundness:** 2 fair
**Presentation:** 2 fair
**Contribution:** 2 fair

**Summary:**

This paper considers two new heterogeneous FL settings, (a) hetero-label FL where the clients have different labeling orders, i.e., the same label space but the ground-truth ids are different permutations across clients, and (b) hetero-domain FL where the clients have disjoint datasets in both instance and label class. To deal with these two challenging scenarios, the authors propose to factorize the model parameters into the product of two vectors plus a sparse bias term, in which the first vector is personalized for a specific client and the second one is shared with all participants. Experiments on CIFAR-10/100 and SVHN datasets with various partition manners show promising results with the proposed method. The studied problem and proposed method are interesting, and the paper has a clear organization. However, I have several concerns about the practicability of the hetero-label FL setting, the writing quality, and the soundness of the paper.


**Questions:**

See questions mentioned in the above weaknesses.

**Limitations:**

IMO, there is no potential negative social impact.


**Strengths And Weaknesses:**

Strengths:
1. The proposed hetero-domain FL setting is novel and practical, which has good potential to extend the application and research scope of FL.
2. The paper has a clear organization and the proposed factorization method is easy to follow.
3. The authors propose to capture client-general and client-specific knowledge via a global-shared base vector and local-personalized coefficient vectors, respectively, which is intuitive and seems to be simple yet effective.

Weaknesses:
1. The practicability of the hetero-label FL setting is questionable. I agree that clients may have different label spaces as different labeling schemes can be adopted. However, the label permutation seems a bit strange and tailored for the simulation, as the clients in realistic applications can easily reorder the labels to make their label orders consistent with negligible computation and communication costs (at the beginning of the FL courses and communicating only once).
Further, if reordering information is forbidden to be exchanged, the users can also leverage privacy-preserving entity resolution techniques to align the label IDs, which are well-studied in vertical FL literature such as [1].

2. The proposed factorization method is simple, while the authors have not given more in-depth analysis and discussions on such topics as (1) The convergence of the proposed method since the converged communication round is also important in FL; (2) The expansibility of the proposed method in terms of the model structures. The authors focus on the fully-connected and convolutional layers. Can the proposed method be applied to other popular and practical NN models containing BN layers, residual structures, attention structures, etc., (3) The space-time complexity analysis with the proposed factorization method. The authors claimed that the proposed method improved FL efficiency multiple times. However, a more formal time-space complexity analysis is lacking in terms of FL training, inference, and communication processes.

3. The experimental datasets should not be applied with the label permutation when examining the performance in the domain-heterogeneous FL setting (see line 265 and paragraph 280). The permutation mixed the effect of label heterogeneity, making the analysis in paragraph 280 of domain heterogeneity confounded. Besides, the number of experimental clients is a bit small (20 clients usually correspond to cross-silo FL). How does the proposed method perform on larger client scales and cross-device FL scenarios?

4. The writing has a lot of room for improvement in terms of preciseness, typos, and grammatical mistakes. For example,
- line 23, "the same set of labels that are in the same order" The elements in a set are *un-ordered*.
- line 43, the numbering of the figures and the order in which the figures appear do not match: Figure 1 appeared after Figure 2
- line 44, the term "instance-wise manner" is un-clear
- line 36 ~ 50. the example is a bit hard to read and it is unclear whether the authors make the classifier layers personalized to each client in the example. Without the classifier layer personalization, the gradients' divergence is obvious and comparison is trivial.
- line 90, downloads -> download
- line 95, use -> using
- line 97, introduces -> introduce
- line 115, initialize -> initializes
- line 127, utilized -> utilize
- line 129, wher -> where
- line 139 and line 149. there are two notations for the client datasets, P_k and D_k. The authors used D_k in equation (6) and how about P_k.
- line 151, the last equation only expresses that there is no instance-wise overlap, while not mentioning the class-wise overlap.
- line 169, uses -> use
- line 177, shows -> show
- equation 1 and equation 2 lack periods
- line 211, another term "task-heterogeneous FL", compared to the term "label-hetero FL", it is better to unify them
- line 213, demonstrated -> demonstrate
- line 214, achieves -> achieve
- line 250, Models -> Baselines,  This section lacks an introduction to the adopted models
- line 257, missing citations for CIFAR-10 and SVHN
- line 259, which labels -> whose labels
- line 331, "As shown, " incomplete sentence
- in appendix Figure 4, there is a reference error: "Table ??"

Reference

[1] Private federated learning on vertically partitioned data via entity resolution and additively homomorphic encryption

---

> ### Author Response · Authors · 2022-07-31
> **Initial Response (3/3) to Reviewer dsaH**
>
>
> **Question 3:** “The experimental datasets should not be applied with the label permutation when examining the performance in the domain-heterogeneous FL setting (see line 265 and paragraph 280). The permutation mixed the effect of label heterogeneity, making the analysis in paragraph 280 of domain heterogeneity confounded. Besides, the number of experimental clients is a bit small (20 clients usually correspond to cross-silo FL). How does the proposed method perform on larger client scales and cross-device FL scenarios?”
>
> **Answer 3:**
> 1. **Additional experiments for domain heterogeneity without label permutation**
>     * As advised, we conducted additional experiments on domain heterogeneous dataset without permuting the labels among the clients in the same domain for more clear analysis. Below is the result.
> |          Model                  | Performance [%] |
> |----------------------------|-----------------|
> | Stand-Alone                | 61.64           |
> | FedAvg                     | 58.08           |
> | Clustered-FL                     | 59.84           |
> | pFedPara                   | 60.33           |
> | **Factorized-FL α (Ours)** | **64.89**       |
> | Factorized-FL β (Ours) |  63.54       |
>
>     * As shown in the table, our methods still largely outperform all baseline models, and we observe that the overall performance is almost identical compared to the previous setting (including label permutation across all clients). We conjecture that the inter-domain knowledge transmission seems to overwhelm whether the labels are permuted or not, which is supported by Figure 2 (a) demonstrating the domain heterogeneity is more significant than the label heterogeneity.
>
> 2. **Additional experiments with 100 clients**
>     * For the cross-device FL scenarios, we use 100 clients for the additional experiments in the domain heterogeneous settings. We follow the the same experimental configurations (domain heterogeneity without label permutation), and we assign 20 clients per domain. Below is the result.
> |          Model                  | Performance [%] |
> |----------------------------|-----------------|
> | Stand-Alone                | 41.05           |
> | FedAvg                     | 39.62           |
> | Clustered-FL                     | 39.77           |
> | pFedPara                   | 39.84           |
> | **Factorized-FL α (Ours)** | **41.76**       |
> | Factorized-FL β (Ours) |  41.55      |
>
>     * As shown in the table, ours still outperforms all baseline models and actually improves performance over the stand-alone model, which tells that our methods can leverage knowledge across domains even with the 100 clients. We observe the performance gap is relatively smaller than that of the experiments with 20 clients. This is because each client learns on 5 times smaller data (250-300 instances per client), but we only train them for 100 rounds due to the lack of time during the rebuttal period. We will include 500-round results in our final revision. Thank you for the constructive feedback.
> ---
>
> **Question 4:** “The writing has a lot of room for improvement in terms of preciseness, typos, and grammatical mistakes”
>
> **Answer 4:** We sincerely thank you for your careful reading. We will reflect all your corrections in the revision. These help a lot in improving the quality of the paper. Particularly, for line 36-50, the classifier layers are not shared when performing federated learning (Figure 2 (c)), as same as our label and domain heterogenous settings. We will clarify it in our revision.

---

> ### Author Response · Authors · 2022-07-31
> **Initial Response (2/3) to Reviewer dsaH**
>
>
> **Question 2:** “More in-depth analysis and discussions on such topics as (1) The convergence of the proposed method since the converged communication round is also important in FL; (2) The expansibility of the proposed method in terms of the model structures. The authors focus on the fully-connected and convolutional layers. Can the proposed method be applied to other popular and practical NN models containing BN layers, residual structures, attention structures, etc., (3) The space-time complexity analysis with the proposed factorization method. The authors claimed that the proposed method improved FL efficiency multiple times. However, a more formal time-space complexity analysis is lacking in terms of FL training, inference, and communication processes.”
>
> **Answer 2:**
> 1. **Comparison of the converged communication round.**
>     * We empirically demonstrated our convergence rate and stability by showing the convergence plots for all experiments averaged over multiple trials (see Figure 6 (a)-(e)).
>     * We additionally provide the converged communication round to directly show communication efficiency. Below is the corresponding results of the domain heterogeneous dataset.
> |                  | Round at Target Acc. 60% | Performance at Round 100 |
> |------------------|----------------------------------------|------------------------------|
> | Stand-Alone      | 56                                     | 61.35                        |
> | Per-FedAvg       | 93                                     | 62.92                        |
> | FedFomo          | 53                                     | 62.07                        |
> | pFedPara         | 82                                     | 61.11                        |
> | Factorized-FL α  | **37**                                 | **64.49**                        |
> | Factorized-FL β  | 63                                     | 63.93                        |
>     * As shown, our methods show significant advantage in communication efficiency as it requires the least communication rounds to reach the target performance compared to the baseline models.
>
> 2. **Expansibility of the proposed methods**
>
>     * We have already shown the applicability of our factorization techniques by combining with the batch normalization and residual connections (please see Figure 7 (k)). Also, we use ResNet architecture as our base network, which leverages Conv, FC, BN, residual connection, etc., and show that they work successfully.
>     * For the attention structures, using the attention based network is beyond our research scope, as we mainly solve image classification problems. However, in our factorization technique, we can consider the task-specialized vector $\textbf{v}$ as the attention-like vector that transforms the task-general vector $\textbf{u}$. As the transformation works successfully, we believe our techniques also can be combined with attention structures.
>
> 3. **The space-time complexity analysis**
>     * We provide the approximated space-time complexity in the table below, compared against those of the essential baselines. Let $W$ be the model size,  $N$ be the number of instances, $H$ be the number of other models (for FedFOMO algorithms), $U$ be the size of the rank-1 vectors (or matrices for pFedPara) in $\mathcal{U}$ , $V$ be that of $\mathcal{V}$, $M$ be the that of $\mathcal{M}$, $\textbf{v}$ be the vector for the second last layer from $\mathcal{V}$.
> |                 | Training                              | Inference                  |  Client-to-Server Cost | Server-to-Client Cost |
> |-----------------|---------------------------------------|----------------------------|----------------------------------|----------------------------------|
> | FedAvg          | $W \cdot N$                           | $W \cdot N$                | $W$                              | $W$                              |
> | pFedPara        | $(U \cdot V)^2 \cdot N$               | $ (U \cdot V)^2 \cdot N$   | $U + V$                          | $U + V$                          |
> | FedFOMO         | $W \cdot N + H \cdot W \cdot N_{val}$ | $W \cdot N$                | $W$                              | $(H+1) \cdot W$                      |
> | Factorized-FL α | $ (U \cdot V + M) \cdot N$            | $ (U \cdot V + M) \cdot N$ | $U + \textbf{v}$                 | $U$                              |
> | Factorized-FL β | $ (U \cdot V + M) \cdot N$            | $ (U \cdot V + M) \cdot N$ | $U + V + M$                      | $U + V + M$                      |
>     * Note that, for communication costs of our methods, $(U \cdot V + M) \approx W$ and thus $(U + V + M) < W$ for both Client-to-Server and Server-to-Client communication costs. Please also refer to the actual size of the data transmission for the communication cost in Table 1 and 2.
>     * For training procedure of FedFOMO, it requires validation step for $H$ number of other model weights on their validation set at each local client.

---

> ### Author Response · Authors · 2022-07-31
> **Initial Response (1/3) to Reviewer dsaH**
>
> #### Thank you for your constructive feedback. We responded to your individual comments below.
>
> ---
>
> **Question 1**: “The practicability of the hetero-label FL setting is questionable. I agree that clients may have different label spaces as different labeling schemes can be adopted. However, the label permutation seems a bit strange and tailored for the simulation, as the clients in realistic applications can easily reorder the labels to make their label orders consistent with negligible computation and communication costs (at the beginning of the FL courses and communicating only once). Further, if reordering information is forbidden to be exchanged, the users can also leverage privacy-preserving entity resolution techniques to align the label IDs, which are well-studied in vertical FL literature such as [1].”
>
> **Answer 1:**
> We would like to emphasize that the label heterogeneity is indeed an important challenge in many of the real-world cross-silo federated learning scenarios we are aiming to tackle, and is not a contrived scenario. Please see the answers below:
> 1. **Permutation of labels cover any kind of label asynchronism**:
>
>     * The label heterogenous scenario we consider does not only consider different “orderings of the labels”, but rather presumes **“different mapping of the labels”**. For example, in the code-level explanation, the labels even annotated in totally different languages are mapped to the same index scale, which is starting from 0 to the number of classes-1, regardless how differently they are labeled. The point is that the mapping of the labels can be different, depending on the labeling schemes at each client, Therefore, the permutation of labels is sufficient to cover any kind of asynchronism, and it does not just present the ordering of the labels.
>
> 2. **The clients cannot easily reorder their labels in the real-world scenarios**
>     * First of all, it significantly violates data privacy for you to share any form of local information and align the labels wherever, which is strictly forbidden in federated learning.
>     * Second, even if such risky behavior is allowed in some strongly limited scenarios, aligning the labels is **still a significantly non-trivial task** if the labels are annotated by totally different labeling schemes, i.e. different languages (English and Japanese) or different codes (disease codes managed by each medical institution, etc).
>     * We are actually considering to apply our model to a real-world cross-silo FL scenario where **disease codes are differently annotated in two different medical institutions**, and it is very difficult to align codes that denote different diseases across the two institutions with two different labeling schemes. In medical fields, there exists experts and organizations that are dedicated to standardizing the database across medical institutions to overcome such issues. Thus, reordering the labels is not an easy problem in the realistic applications as argued.
>     * We updated our paper so that readers can better understand the limitations of label alignment in real-world applications. Thank you for your feedback.
>
> 3. **Vertical FL methods cannot be directly applied to our tasks**
>     * We tackle multi-class classification tasks on **3-dimensional unstructured data**, which is orthogonal to the task you mentioned that deals with 1-dimensional structured data, which mainly focus on (logistic) regression tasks [1].
>     * Moreover, while the 1-dim structured data can be divided in a feature-wise manner and distributed to each client (Vertical FL), unstructured data like images is **not generally handled in such a manner**, since, when we split or decompose them into some slices, the sliced images might become not human-recognizable anymore.
>     * See [2,3] for their Vertical FL experiments on image dataset, i.e. MNIST, CIFAR-10, etc., and how their setup is highly “artificial” and “unrealistic”. They divided the images into a bunch of slices to perform Vertical FL, which made the images not human-readable anymore, and distributed them to the clients. We doubt how many such cases actually exist in real-world applications.
>
> [2] Xia et al, A Vertical Federated Learning Framework for Horizontally Partitioned Labels, 2021
>
> [3] Fu et al, Label Inference Attacks Against Vertical Federated Learning, 2022

---

> ### Comment · Reviewer_dsaH · 2022-08-08
> **Thanks for the response.**
>
> I thank the author for making additional experiments and providing detailed responses. These responses partially address my questions, and I am willing to raise my rating from 3 to 4 due to some of the following considerations that remain.
>
> 1. The authors have re-emphasized their primary focus on the cross-silo scenario, but the current version is still somewhat unclear as to the positioning of the paper. As shown in response A1, and also in response A3, the experimental results in the cross-device scenario are not significant. On the one hand, it is not trained through (100/500 rounds), on the other hand, the FedAvg in this result is much worse than the stand-alone mode, and the proposed method is only slightly better than the stand-alone mode, I think the experiment needs more rigorous implementation and checking.
>
> 2. If the results mentioned in A3.1 are almost identical to the overall performance already available, doesn't that just show that it is doubtful that the heterogeneous tagging setup is necessary in this simulation. Other reviewers, such as BK7j have also raised similar questions.
>
> 3. The revision version will have considerable changes, both in the writing and in the main experiment. Although the authors did not upload their final revision version, I can understand this due to time constraints.

---

> > ### Author Response · Authors · 2022-08-09
> > **Author Response to Reviewer dsaH**
> >
> > Thank you for getting back to us and raising your score. We responded to your remaining concerns below:
> >
> > ---
> >
> > **Comment 1**: “The authors have re-emphasized their primary focus on the cross-silo scenario, but the current version is still somewhat unclear as to the positioning of the paper. As shown in response A1, and also in response A3, the experimental results in the cross-device scenario are not significant. On the one hand, it is not trained through (100/500 rounds), on the other hand, the FedAvg in this result is much worse than the stand-alone mode, and the proposed method is only slightly better than the stand-alone mode, I think the experiment needs more rigorous implementation and checking.”
> >
> > **Answer 1**:
> >
> > 1. **For the 100-client experiment, it is natural that the stand-alone model shows better performance over FedAvg model in the domain heterogenous scenario**, since significant knowledge collapse happens when performing uniform knowledge aggregation across extremely heterogeneous domains. Not only for the FedAvg, but also all other baseline models fail to properly leverage the other clients’ knowledge, which even deteriorates their local knowledge. This is why the stand-alone model outperforms them. Besides, **this phenomenon has been already observed and discussed in the previous experiments (Table 1 Bottom of the main paper)**, in other words, **the number of clients is not a significant factor for analyzing the domain/label heterogeneous scenarios.** It is more related to adaptation in the few-shot learning scenarios (around 25 instances per class). With the sufficient amount of round, we expect that our method further outperforms other models with large margins, as same as the Table 1 experiment. We ensure that we will include a 500-round experiment in our final revision (it requires significant time to run multiple trials for all baseline models).
> >
> > 2. **Our method works for both cross-silo and cross-device federated learning scenarios.** As our method can be applied to both scenarios, there is no need to choose/emphasize one particular scenario among them. In fact, there are many federated learning studies that do not fall into one scenario, since most of them can be applied to both cross-silo/device scenarios. **Please see the recent FL works we cite [4,5,6] do not mention whether they focus on either cross-silo or cross-device FL**, but they rather discuss whether their method tackles either the standard federated learning or personalized federated learning scenarios.
> >
> > [4] Nam et al, FedPara: Low-Rank Hadamard Product for Communication-Efficient Federated Learning, ICLR 2022
> >
> > [5] Zhang et al, Personalized Federated Learning with First Order Model Optimization, ICLR 2021
> >
> > [6] Sattler et al, Clustered Federated Learning: Model-Agnostic Distributed Multi-Task Optimization under Privacy Constraints, IEEE NNLS 2020.
> >
> > ---
> >
> > **Comment 2**: “If the results mentioned in A3.1 are almost identical to the overall performance already available, doesn't that just show that it is doubtful that the heterogeneous tagging setup is necessary in this simulation. Other reviewers, such as BK7j have also raised similar questions.”
> >
> > **Answer 2**: We thank you for your constructive suggestion. **We will introduce the pure domain heterogeneous setting in the main document** and move the previous experiment (Table 1 Bottom) to the Appendix as the additional analysis.
> >
> > ---
> >
> > **Comment 3**: “The revision version will have considerable changes, both in the writing and in the main experiment. Although the authors did not upload their final revision version, I can understand this due to time constraints.”
> >
> > **Answer 3**: Thank you for your understanding. As you mentioned, there will be considerable changes in the revision (i.e. **correcting writing errors, moving Table 1 Bottom to Appendix and adding a new table for the pure domain heterogeneous setting in the main document, adding result and analysis for 100-client experiment with 500 rounds, etc.**) As some of the new experiments require statistics from the multiple trials, we genuinely appreciate your thoughtful understanding and we will update the revision as soon as possible. Thank you again for your dedicating effort to enhance our paper.

---

### Official Review · Reviewer_BK7j · 2022-07-10

**Rating:** 7
**Confidence:** 3
**Soundness:** 3 good
**Presentation:** 3 good
**Contribution:** 3 good

**Summary:**

This paper introduces a personalized federated learning method based on parameter factorization and similarity matching. The authors factorize the model parameters into a pair of vectors to condense common and personalized knowledge, which are used for guiding the model update aggregation. Extensive experiments show that the proposed method can better handle both label and domain heterogeneity of data in federated learning than many existing methods.

**Questions:**

N/A

**Limitations:**

The authors have adequately addressed the limitations and potential negative societal impact.

**Strengths And Weaknesses:**

Strengths:
1. This paper studies a setting with stronger data heterogeneity.
2. The proposed method is interesting.
3. The experimental evaluation is solid.
4. This paper is well written.

Weakness:
I only have one concern on the experimental setting. The authors group the 100 classes to simulate the domain heterogeneous scenario, but this setting is quite similar to the standard extreme label heterogeneous scenario where each client only keeps one class of samples. Thus, the boundary between label and domain heterogeneous scenarios are somewhat vague. I think a simplified scenario where different clients keep different domains of data but their label space is similar (a good example is review sentiment analysis), is also interesting.

---

> ### Author Response · Authors · 2022-07-31
> **Initial Response to Reviewer BK7j**
>
> #### We thank you for your positive comments and constructive feedback. Please see the additional experiment below. We will include the analysis in our revised paper.
> ---
> **Suggestion:** “The boundary between label and domain heterogeneous scenarios are somewhat vague. I think a simplified scenario where different clients keep different domains of data but their label space is similar (a good example is review sentiment analysis), is also interesting.”
>
> **Answer:** Thank you for your constructive suggestion. We agree that decoupling the domain and label heterogeneity improve the clarity when analyzing the effectiveness of our methods. As advised, we conduct additional experiments for domain heterogenous settings without label permutation.
> |          Model                  | Performance [%] |
> |----------------------------|-----------------|
> | Stand-Alone                | 61.64           |
> | FedAvg                     | 58.08           |
> | Clustered-FL                     | 59.84           |
> | pFedPara                   | 60.33           |
> | **Factorized-FL α (Ours)** | **64.89**       |
> | Factorized-FL β (Ours) |  63.54       |
>
> As shown in the table, our methods still largely outperform all baseline models and we observe the overall performance is almost identical compared to the previous setting (including label permutation across all clients). We conjecture that the inter-domain knowledge transmission seems to overwhelm whether the labels are permuted or not, which is supported by Figure 2 (a) demonstrating the domain heterogeneity is more significant than the label heterogeneity.

---

### Official Review · Reviewer_7Csy · 2022-07-11

**Rating:** 6
**Confidence:** 3
**Soundness:** 3 good
**Presentation:** 3 good
**Contribution:** 3 good

**Summary:**

This paper introduces a new Agnostic Personalized Federated Learning (APFL) problem and study the personalized federated learning method with parameter factorization and similarity matching.  Specifically, the authors focus on label heterogenous and domain heterogeneous problems.  All in all, the proposed method shows good performance compared to other personalized federated learning baselines.

**Questions:**

Some of my concerns/suggestions for improvements:

1. Could you please elaborate more on why rank-1 vectors with sparse matrix? What if we consider other values of rank gamma != 1 with local bias (Figure 3 left)? I did not see (or may miss) the ablation study of different rank values, and perhaps discussions of trade-offs for rank > 1. I would love to see more analyses rather than only Figure 7 (h). (Same for Figure 7 (i))
2. In line 46-50, could you please elaborate more on why permuting the labels and learning on different datasets make gradients more diverge?
3. In Figure 2 (c), if we run for more epochs, what are the performance? Will it still look ‘consistent’ as in the current version?
4. Why Factorized-FL alpha and Factorized-FL beta performs ‘inconsistently’ compared to each other in Table 1 page 7 w.r.t both Label-heterogeneous and Domain-heterogeneous?
5. [Minor] Figure 7 (a)-(e) are hard to see.



**Limitations:**

The authors did have clear discussions about limitations and potential negative societal impact in Section A in the Appendix.

**Strengths And Weaknesses:**

Some strengths:

1. The paper tackles an interesting problem with personalized federated learning. The paper has very extensive experiments with great analyses. The contributions are good.
2. According to Table 1 on page 7, Factorized-FL (both versions) shows good results across datasets and scenarios (domain and label heterogeneous)
3. It’s also good that the authors provided extensive experiments on the Appendix as well as source code for reproducibility.


Some of my concerns/suggestions for improvements:

1. Could you please elaborate more on why rank-1 vectors with sparse matrix? What if we consider other values of rank gamma != 1 with local bias (Figure 3 left)? I did not see (or may miss) the ablation study of different rank values, and perhaps discussions of trade-offs for rank > 1. I would love to see more analyses rather than only Figure 7 (h). (Same for Figure 7 (i))
2. In line 46-50, could you please elaborate more on why permuting the labels and learning on different datasets make gradients more diverge?
3. In Figure 2 (c), if we run for more epochs, what are the performance? Will it still look ‘consistent’ as in the current version?
4. Why Factorized-FL alpha and Factorized-FL beta performs ‘inconsistently’ compared to each other in Table 1 page 7 w.r.t both Label-heterogeneous and Domain-heterogeneous?
5. [Minor] Figure 7 (a)-(e) are hard to see.


Overall, the paper is good in my opinion.

---

> ### Author Response · Authors · 2022-07-31
> **Initial Response (2/2) to Reviewer 7Csy**
>
> **Question 3:** “In Figure 2 (c), if we run for more epochs, what are the performance? Will it still look ‘consistent’ as in the current version?”
>
> **Answer 3:**
> * We additionally conduct experiments for 5/10/20 epochs on domain heterogeneous datasets for 100 rounds.
> |              Epochs              | 5        | 10       | 20       |
> |----------------------------|-----------------|-----------------|-----------------|
> | Stand-Alone                | 61.35           | 62.69           | 63.91           |
> | FedAvg                     | 56.42           | 62.27           | 65.38           |
> | pFedPara                   | 61.11           | 63.30           | 65.76           |
> | FedFOMO                    | 62.07           | 63.46           | 64.30           |
> | **Factorized-FL α (Ours)** | **64.49**       | **65.58**       | 66.73       |
> | **Factorized-FL β (Ours)** | 63.93       | 65.28       | **69.14**       |
> * As shown in the table, we observe that the overall performance increases as the local epoch increases and our methods consistently show better performance compared to the base models with larger local epochs. Interestingly, Factorized-FL $\beta$ outperforms Factorized-FL $\alpha$ when the local epoch is 20. We conjecture that the sufficient amount of local epochs improves the similarity matching for clients in the same domain to be more correctly identified and encourages task-specific knowledge $\mathcal{V}$ and $\mathcal{M}$ to be shared between them, which further improves the performance. Thank you for the insightful feedback. We will include this in our final revision.
>
> ---
>
> **Question 4:** “Why Factorized-FL alpha and Factorized-FL beta performs ‘inconsistently’ compared to each other in Table 1 page 7 w.r.t both Label-heterogeneous and Domain-heterogeneous?”
>
> **Answer 4:**
>
> * **For the label heterogeneous scenario, Factorized-FL $\beta$ consistently outperforms over Factorized-FL $\alpha$**. This is because, Factroized-FL $\alpha$ uses only 2% of weight sharing compared to Factorized-FL $\beta$, as well as Factorized-FL $\beta$ effectively alleviates knowledge collapse often caused by the element-wise aggregation in high dimensional weight space by reducing parameter space into the vector spaces. On the other hand, Factorized-FL $\alpha$ shows **consistent performance regardless of whether the labels are permuted**, which most of the other models including Factorized-FL $\beta$ are degenerated when the labels are permuted.
>
> * **For the domain heterogeneous scenario, Factorized-FL $\alpha$, even with 2% of weight sharing, shows almost identical or even better performance, and finally outperforms Factorized-FL beta on average.** When the domain is extremely heterogeneous, sharing and merging the client-specific knowledge deteriorates the model performance, as the knowledge is not beneficial to each other. Thus, Factorized-FL $\alpha$ which only shares the domain-general knowledge outperforms all other models including Factorized-FL $\beta$.
>
> ---
>
>
> **Question 5:** "[Minor] Figure 7 (a)-(e) are hard to see."
>
> **Answer 5:** Thank you for your constructive feedback. We will revise the ratio of the figure to improve readability in the revision.

---

> ### Author Response · Authors · 2022-07-31
> **Initial Response (1/2) to Reviewer 7Csy**
>
> #### We thank you for your constructive comments. We carefully responded to your feedback and will reflect them in our revision.
> ---
> **Question 1:** “Could you please elaborate more on why rank-1 vectors with sparse matrix? What if we consider other values of rank gamma != 1 with local bias (Figure 3 left)? I did not see (or may miss) the ablation study of different rank values, and perhaps discussions of trade-offs for rank > 1. I would love to see more analyses rather than only Figure 7 (h). (Same for Figure 7 (i))”
>
> **Answer 1**
> 1. **We additionally conduct experiments with new baseline model - (1) Low rank matrix and (2) Low rank matrix with local bias.**
>     * We use the same domain heterogeneous dataset while varying the gamma values [10, 100, 200].
> |                                         | $\gamma$ | Performance [%] |
> |-----------------------------------------|----------|-----------------|
> | Stand-Alone                             | -        | 61.35           |
> | FedAvg                                  | -        | 56.42           |
> | Low Rank Matrix with Bias $\mathcal{M}$ | 10       | 63.27           |
> | Low Rank Matrix with Bias $\mathcal{M}$ | 100      | 57.51           |
> | Low Rank Matrix with Bias $\mathcal{M}$ | 200      | 56.85           |
> | **Factorized-FL α (Ours)**              | 1        | **64.49**       |
> | Factorized-FL β (Ours)              | 1        | 63.93       |
>     * As shown in the table, ours outperforms all base models. This is because such weight matrices higher than rank 1 may result in larger coordinate-wise incompatibility of the parameters. We can observe that, the larger the rank, the closer the performance is to Fedavg. As the sufficiently high rank, which is not much different from the original model parameters, could lead to knowledge collapse more frequently and easily when performing aggregation.
> 2. **$\tau$ ensures to find the helpful weights that their similarity scores are above a certain degree.**
>     * Thus the lower $\tau$ allows less similar model weights more frequently when performing similarity matching and aggregation. In Figure 7 (h), the lower $\tau$s show gradual performance degenerations. This demonstrates that allowing the unhelpful knowledge more frequently affects the model performance and $\tau$ successfully controls it. We observe that the adequate value for the $\tau$ is in the range of [0.85 - 0.95] depending on the tasks.
>
>  ---
> **Question 2**: “In line 46-50, could you please elaborate more on why permuting the labels and learning on different datasets make gradients more diverge?”
>
> **Answer 2:** We conjecture that the models are more likely to learn a different order of convolutional filters and neurons for the fully connected layers when the labels are permuted on the same data. In Figure 4 (a), we have shown that the distance between the coefficient knowledge $\textbf{v}$s is larger than that between the base knowledge $\textbf{u}$s in the label heterogeneous scenario. As illustrated in Figure 5 (a) 1, $\textbf{v}$s capture the filter-specific knowledge, which may learn the different order of filters, it may transform the filter base knowledge $\textbf{u}$ (2) to reconstruct the full weights (3) in different orders.

---

### Official Review · Reviewer_fR1b · 2022-07-12

**Rating:** 6
**Confidence:** 3
**Soundness:** 3 good
**Presentation:** 4 excellent
**Contribution:** 2 fair

**Summary:**

In this paper, the authors study a new and interesting problem, i.e., they consider the heterogeneity of both the labels and tasks in personalized federated learning, which is very different from the assumptions in previous works, i.e., a same set of labels and a same task in different clients. The studied problem, termed as the Agnostic Personalized Federated Learning (APFL) problem, is well motivated, and the associated challenges are well identified.

Then the authors then propose a novel solution, i.e., Factorized-FL, for the studied problem. Specifically, Factorized-FL factorizes model parameters to reduce the parameter in order to alleviate the knowledge collapse, and utilizes them to measure the task-level similarity for matching relevant clients. Empirical studies on public datasets also show the effectiveness and efficiency of Factorized-FL.


**Questions:**

1 Is there some simple approach to address the label heterogeneity challenge, e.g., label alignment before training?

2 In federated learning, privacy protection is an important issue. In particular, in the similarity matching step, the calculation of similarity requires V of other clients, which may disclose the privacy of the clients. However, the authors do not discuss about the privacy issue. Is there some theoretical guarantee about the privacy protection?


**Limitations:**

The authors are suggested to include more discussions or theoretical results on privacy protection.

**Strengths And Weaknesses:**

Originality:
It is an interesting and novel work though the technique is relatively simple.

Quality:
The main idea is simple, i.e., factorization of the parameter into a sharing part and a non-sharing part.

Clarity:
The paper is well presented with good illustrations and explanations.

Significance:
It is a nice work which addresses an important problem, i.e., heterogeneity in labels and tasks, and achieves significant improvement over the SOTA methods.

---

> ### Author Response · Authors · 2022-07-31
> **Initial Response to Reviewer fR1b**
>
> #### We sincerely thank you for your constructive feedback. We carefully responded to all your comments and will reflect them in our revision.
>
> ---
>
> **Question 1:** “Is there some simple approach to address the label heterogeneity challenge, e.g., label alignment before training?”
>
> **Answer 1:**  It is a non-trivial task to align the labels without sharing the local data directly.
> * The simplest way is to share the local data at the beginning of the federated learning and align their labels across all local data at server and distribute them back to the clients. However, sharing the local data with others significantly violates protection of the data privacy and it is **strictly prohibited** in federated learning
> * Even If it is allowed in some strongly limited scenarios, it is still impractical to synchronize the labels when the labels are annotated by totally different labeling schemes, i.e. different languages (English and Japanese) or different codes (disease codes managed by each medical institution, etc).
> * We are actually considering to apply our model to a real-world cross-silo FL scenario where disease codes are differently annotated in two different medical institutions, and it is very difficult to align the disease codes across the two institutions with two different labeling schemes. In medical fields, there exists experts and organizations that are dedicated to standardizing the database across medical institutions to overcome such issues.
> * In sum, aligning the labels annotated by different label schemes is not an easy problem and should be handled carefully as it shows a negative impact on model training in federated learning.
>
> ---
>
>
> **Question 2:** “The calculation of similarity requires V of other clients, which may disclose the privacy of the clients, Is there some theoretical guarantee about the privacy protection?”
>
> **Answer 2:**  Although there is no theoretical guarantee, It is more difficult to recover the local distribution only using a vector $v^{L-1}_k$, compared to using the full weights.
> * As you know, most of federated learning algorithms send their full model weights (or gradients) to the server by default, i.e. FedAvg. We consider sharing the model weights as safer than sharing local data directly, as it is hard to recover the local data distribution from the model parameters.
> * FedFOMO, one of our baseline models, even downloads a few model weights of other clients from the server and validates the helpfulness of them at each client, but it is still considered as safe since it at least does not share local data directly.
> * For our method, in this context, we believe that using a single factorized vector $v^{L-1}_k$ is less risky than utilizing full model weights. Moreover, $v^{L-1}_k$ is a vector factorized even only from the second last layer, which is a super small part of the entire network. Also, without $\mathcal{U}$ and bias mask $\mathcal{M}$, it is even impossible to construct the original model weights before saying the recovery of the original local data distribution from the model weights.
> * Besides, we discuss in the limitation section (Section A of Appendix) that such privacy-protection issues when transmitting model weights are rather open research problems which are essential and crucial for the current federated learning algorithms.

---

### Author Response · Authors · 2022-08-03
**Initial Author Response**

Dear Reviewers,

We sincerely thank you for your time and efforts in reviewing our paper as well as the constructive and helpful feedback. We carefully read and responded to all your comments. During this author rebuttal period, we have conducted following additional experiments and analysis:

 * We have conducted experiments on domain heterogeneous dataset without label permutation.
 * We have conducted experiments with a larger number of clients.
 * We have conducted experiments with a larger number of epochs.
 * We have conducted experiments with a new baseline model, namely Low Rank Matrix with Bias Matrices.
 * We have provided in-depth analysis and detailed explanations to address concerns from the reviewers.

Thanks to your helpful suggestions, we believe that our paper has been largely improved and has become stronger with the additional experimental results and analysis. We will include all the feedback in our revised paper.

Thank you,
Authors

---

### Meta-Review · Area_Chair_mv4Q · 2022-08-26

**Recommendation:** Accept
**Confidence:** Less certain

**Metareview:**

In this paper, the authors study an interesting problem where clients are heterogeneous in both labels and learning tasks/domains. This problem is different from most of the existing pFL settings, and has great potential to broaden the application of FL. To handle this case, the authors propose a novel method based on parameter factorization and similarity matching. Experimental results on both label and domain heterogeneous cases are promising. To this end, I recommend accepting this submission.

However, I do have some concerns regarding the experimental setting of the heterogeneous domain (as pointed out by reviewer BK7j and dsaH). Without proper dataset simulation, it makes the experimental results less convincing.

This submission can be further improved from all the discussions between reviewers and the authors. Hope they find the discussion useful and make this submission a better one


**Award:**

No

---

### Decision · Program_Chairs · 2022-09-14

Accept